# PHYSICAL DYNAMICS AS NEXT GEOMETRIC GRAPH PREDICTION

## ABSTRACT

Physical dynamics simulation serves as a foundational component in scientific computing and AI applications. This paper presents a novel approach that redefines the problem as autoregressive prediction of spatiotemporal graph sequences. Built upon the expressivity of Transformer, we propose an Equivariant Spatiotemporal Transformer (EST), extending conventional Transformers with specialized equivariant spatiotemporal blocks. These blocks systematically alternate between spatial and temporal modules, rigorously maintaining E(3) symmetries throughout the process. Moreover, the design incorporates a novel Temporal Difference Graph (TDG) module derived from frame-wise variations, effectively modeling global dynamics and addressing cumulative errors in autoregressive predictions. Unlike traditional graph neural networks, our EST can process variable-length historical sequences and mitigate the persistent challenge of error accumulation in autoregressive processes. Comprehensive evaluations across multiscale physical systems (molecular-, protein-, and macroscopic-scale) demonstrate that our method achieves state-of-the-art performance, thereby showcasing its robust and versatile dynamics simulation capabilities.

## 1 INTRODUCTION

Accurately simulating the dynamics of physical systems forms the cornerstone of numerous applications. For example, in drug discovery, molecular dynamics simulations provide profound insights into the binding interactions between drug molecules and their target proteins (Salo-Ahen et al., 2020). Plenty of methods (Battaglia et al., 2016; Sanchez-Gonzalez et al., 2020) have emerged to simulate physical dynamics as graph translation via Graph Neural Networks (GNNs), given that many physical systems can be effectively represented as graphs. Further advancements have been made by leveraging geometric GNNs (Fuchs et al., 2020; Huang et al., 2022), which ensure the dynamics to be independent to any rotation, reflection, or translation transformations, thereby aligning seamlessly with E(3) symmetries inherent in physics. Building upon geometric GNNs, several studies (Xu et al., 2023; Wu et al., 2024) adopt a spatiotemporal approach, rather than the previous frame-to-frame setting, which leverages multiple frames to predict the next one, thereby capturing historical information and recovering non-Markovian interactions.

In recent years, the Transformer architecture (Vaswani, 2017) along with its next-token prediction fashion have become the de facto standard in Natural Language Processing (NLP) and various other domains. Nevertheless, its application in physical dynamics simulations—particularly in graph-based contexts—remains underexplored. Given that both natural languages and physical trajectories are sequential data, it is promising to leverage the success of Transformers for physical dynamics generation. However, notable differences exist between the two tasks. From a data structure perspective, we need to process spatiotemporal graphs rather than text sequences, and the objective shifts from next-token prediction to next-graph prediction. Importantly, the model must conform to certain physical rules, such as E(3) symmetries, to ensure generalizability across arbitrary coordinate systems. In addition, addressing cumulative errors is crucial for autoregressive simulation, necessitating specific design considerations for the model. Consequently, it is infeasible to directly apply standard Transformer (Vaswani, 2017) for physical dynamics simulation.

To effectively bridge these gaps, we propose Equivariant Spatiotemporal Transformers (EST). By inheriting the encoder-decoder architecture from the original Transformer, EST can accept spatiotem-

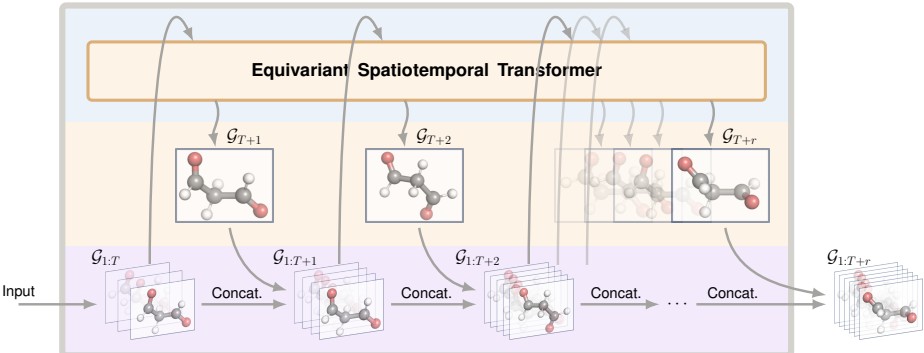

Figure 1: Illustration of how Equivariant Spatiotemporal Transformers (EST) work on molecular dynamics. It processes full spatiotemporal inputs and predict future frames autoregressively.

poral inputs of arbitrary temporal length and predict future graphs in an autoregressive manner, as illustrated in Fig. 1. The core of EST consists of E(3)-equivariant spatiotemporal blocks, which alternate between spatial and temporal modules while preserving E(3) symmetries. More importantly, we provide a novel and effective solution to circumvent cumulative errors, which is a well-known yet underexplored challenge in dynamics simulation. Specifically, we predict the difference from the last frame rather than predicting the next frame directly, as estimating the difference between the current frame and the next may be easier than making a direct prediction of the next frame. To achieve this, we introduce a Temporal Difference Graph (TDG). Initialized as the difference between the last two frames in the input layer, the TDG interacts with all other frames to gather global dynamical patterns in the following layers and and is used alongside the last frame for next-frame prediction. The encoder takes as input the initial trajectory and all predicted frames, while the decoder's input is further augmented with the TDG.

In contrast to existing geometric GNN-based methods (Xu et al., 2023; Wu et al., 2024) that usually assume fixed-length input and output setting, EST exhibits several crucial benefits by inheriting strong expressivity and flexibility of Transformer. In general, our method offers two key advantages: (1) The temporal attention mechanism effectively captures underlying dynamics of variable-length historical trajectories; (2) The autoregressive prediction, now a dominant paradigm in generative AI, naturally aligns with the sequential nature of physical trajectories. Additionally, while E(n) symmetries are well-studied, EST uniquely integrates them into Transformer's encoder-decoder design.

In summary, the contributions of this paper can be summarized as follows:

- We propose EST, a novel Transformer to simulate physical dynamics autoregressively. EST inherits the strong expressivity and flexible designs from original Transformers, while promisingly respecting the spatiotemporal geometries and E(3) symmetries.

- We define TDG to reduce the impact of cumulative errors in rollouts. Initialized as the difference between the last two frames, the TDG interacts with all other frames to capture global dynamics and is used alongside the last frame for next-frame prediction.

- We conduct extensive experiments on real-world datasets across three different scales: molecules, proteins, and human motions. Results demonstrate that our model achieves superior performance across various challenging settings and exhibits a larger gap in longer-horizon rollout tasks compared to existing State-Of-The-Art (SOTA) models.

## 2 RELATED WORK

**Physical Dynamics Simulation.** The simulation of physical dynamics has gained significant attention due to its real-world applications. The Interaction Network (IN) (Battaglia et al., 2016) introduced GNN-like message passing to model object interactions, inspiring researchers to leverage GNNs for their ability to capture complex relational structures (Sanchez-Gonzalez et al., 2019; Li et al., 2020).

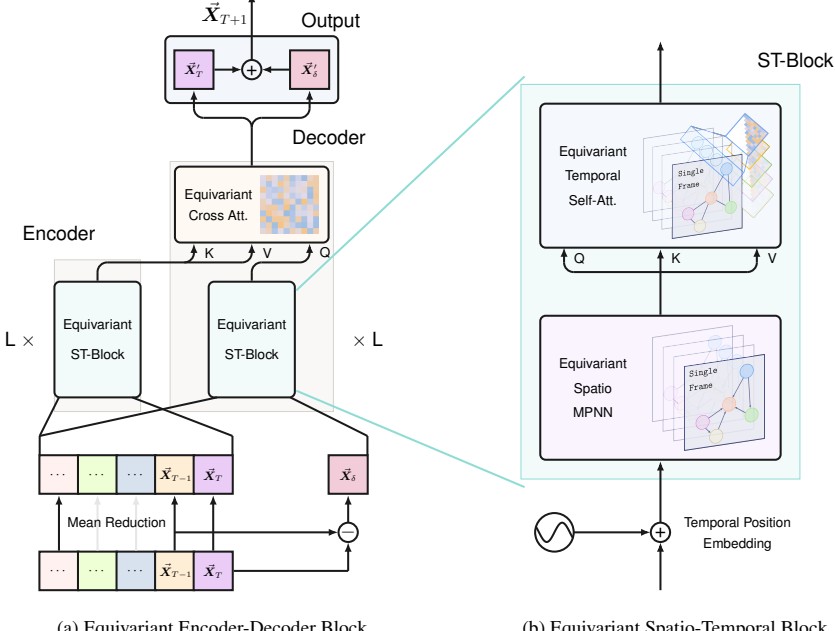

(a) Equivariant Encoder-Decoder Block      (b) Equivariant Spatio-Temporal Block

Figure 2: The overall architecture of EST. Initially, a "Mean Reduction" operation is applied to the historical trajectory $\vec{X}_1, ..., \vec{X}_T$. The entire historical trajectory is then input into an equivariant encoder. After initializing $\vec{X}_\delta$ with the last two frames, we concatenate the historical trajectory with $\vec{X}_\delta$ to form the input for the equivariant decoder. Temporal and spatial dependencies are captured using Equivariant ST-Blocks and Equivariant Cross-attention modules. The final updated $\vec{X}'_T$ is summed with $\vec{X}'_\delta$ to generate the coordinates for the subsequent frame, $\vec{X}_{T+1}$.

However, many methods overlooked physical symmetries, prompting the development of equivariant GNNs that encode geometric information (Thomas et al., 2018; Fuchs et al., 2020; Huang et al., 2022). Despite these advancements, existing approaches often face limitations such as fixed-length inputs (Xu et al., 2023; Wu et al., 2024), single-frame inputs (Satorras et al., 2021), or a focus on single-frame predictions (Han et al., 2022b). In contrast, our EST is the first Transformer-based method for physical simulation that preserves E(3) symmetries while handling both variable and fixed-length inputs—enabling autoregressive trajectory prediction and addressing a critical gap in geometric deep learning.

**Deep Spatiotemporal Models.** Deep spatiotemporal models have gained prominence across diverse real-world applications (Cini et al., 2024; Marisca et al., 2024). In traffic prediction, STGCN (Yu et al., 2017) and DCRNN (Li et al., 2017) leverage graph structures with recurrent or convolutional layers. Transformer-based models such as MMST-ViT (Lin et al., 2023) and MOIRAI (Woo et al., 2024) have been applied to various spatiotemporal tasks including crop yield prediction and general time-series analysis. While effective in their domains, these models typically lack consideration for physical symmetry, limiting their applicability to dynamic simulation. To address this limitation, we adapt the Transformer architecture for 3D physical dynamics simulation by introducing equivariant designs that conform to E(3) symmetries, effectively bridging the gap between general spatiotemporal modeling and physical dynamics simulation.

## 3 OUR METHOD

In this section, we first introduce the necessary preliminaries related to physical dynamics simulation. Then, we describe the framework of our model, which adopts an equivariant encoder-decoder architecture, along with a temporal difference graph to reduce the cumulative errors in autoregressive trajectory predictions. Fig. 2 illustrates the overall framework of our model.

### 3.1 Notations and Definitions

The trajectory of a physical system (*e.g.* a molecule) over a temporal length $T$ and with a time lag of $\Delta t = 1$[1], can be modeled as a spatiotemporal graph $\mathcal{G}_{1:T} := \{\mathcal{G}_t = (\mathcal{V}_t, \mathcal{E})\}_{t=1}^T$, where different frame $\mathcal{G}_t$ shares the same node identities (*e.g.* atoms) and edge connections (*e.g.* bonds), and the $i$-th node $v_{t,i}$ at time $t$ is associated with an invariant feature $\boldsymbol{h}_{t,i} \in \mathbb{R}^c$ (*e.g.* atom types) and an equivariant 3D coordinate vector $\vec{\boldsymbol{x}}_{t,i} \in \mathbb{R}^3$. Particularly for $\boldsymbol{h}_t$, we further add temporal position embedding with the sine function. We do not leverage edge features in this work. We hereafter denote by the matrices $\boldsymbol{H}_t$ and $\vec{\boldsymbol{X}}_t$ the collection of all node features and coordinates in $\mathcal{G}_t$.

**Task Formulation.** As illustrated in Fig. 1, given the observed trajectory $\mathcal{G}_{1:T}$, our goal is to learn a function $\phi$ that autoregressively predicts the future frames $G_{T+1:T+R}$ over a duration of $R \gg 1$. This process can be formally expressed as: $\mathcal{G}_{T+1} = \phi(\mathcal{G}_{1:T}), \dots, \mathcal{G}_{T+r} = \phi(\mathcal{G}_{1:T+r-1})$, where $r = 1, 2, \dots, R$. The above autoregressive prediction is also called rollout process in the domain. In practice, we only require to predict 3D coordinates $\vec{\boldsymbol{X}}_{T+r}$, while the corresponding invariant features $\boldsymbol{h}_{T+r}$ can be computed manually.

**Equivariance.** An important inductive bias to consider is that the function $\phi$ should be E(3)-equivariant, ensuring that the dynamics remain independent of the observation perspective. This means that if the input trajectory undergoes any arbitrary translation, reflection, or rotation transformation, the output of $\phi$ should undergo a corresponding transformation. Further discussion is provided in § B.11.

**Comparisons with Previous Settings.** One remarkable benefit of our function defined above lies in its flexibility to handle inputs of variable temporal lengths. It can take all the historical frames as input for modeling the complete context as defined by default; in case of efficiency considerations, it can also be modified to accept only fixed-length inputs, which degenerates to $\mathcal{G}_{T+r} = \phi(\mathcal{G}_{T+r-L:T+r-1})$ using only previous $L$ frames to predict the next frame. On the contrary, previous methods such as EGNN (Satorras et al., 2021), GNS (Sanchez-Gonzalez et al., 2020), and ESTAG (Wu et al., 2024) can only admit the fixed-length input setting. Furthermore, our task focuses on autoregressive simulation of dynamics, whereas existing methods mostly focus on predicting only one future frame. The architecture of $\phi$ is specifically designed and will be elaborated on in the following subsections.

### 3.2 The Proposed EST Model

Inspired by the original Transformer framework (Vaswani, 2017), the architecture of our proposed EST also consists of an encoder and a decoder, as displayed in Fig. 2. To predict $G_{T+r}$, the encoder takes as input the initial trajectory and all predicted frames, namely $\mathcal{G}_{1:T+r-1}$. The decoder's input is further augmented with an artificial graph $\mathcal{G}_\delta$, forming the input $(\mathcal{G}_{1:T+r-1}, \mathcal{G}_\delta)$. Both the encoder and the decoder employ a certain number of equivariant spatiotemporal blocks for representation learning. The output from the encoder is integrated into the decoder through an equivariant cross-attention layer. Further details are provided below.

(1) **Equivariant Encoder**

The encoder is comprised of $L$ equivariant spatiotemporal blocks. For each block, it alternates two modules: equivariant spatial message passing and equivariant temporal self-attention passing. We denote the $t$-th frame of the $l$-th layer as $\mathcal{G}_t^{e,l}(\boldsymbol{h}_t^{e,l}, \vec{\boldsymbol{X}}_t^{e,l})$. Diagram for ST-Block is shown in Fig. 6(b).

The spatial module leverages EGNN (Satorras et al., 2021) to characterize the spatial geometry of each frame individually, which is formally delineated in Eq. (1). We provide a comprehensive justification for why selecting EGNN as our backbone in § B.3.

$$\boldsymbol{h}_{t,i}^{e,l+1} = \boldsymbol{h}_{t,i}^{e,l} + \varphi_h\left(\boldsymbol{h}_{t,i}^{e,l}, \sum_{j \in \mathcal{N}(i)} \boldsymbol{m}_{t,j}^{e,l}\right), \vec{\boldsymbol{x}}_{t,i}^{e,l+1} = \vec{\boldsymbol{x}}_{t,i}^{e,l} + \frac{1}{|\mathcal{N}(i)|} \sum_{j \in \mathcal{N}(i)} \varphi_x(\boldsymbol{m}_{t,ij}^{e,l}) \cdot (\vec{\boldsymbol{x}}_{t,i}^{e,l} - \vec{\boldsymbol{x}}_{t,j}^{e,l}),$$

(1)

---

[1]Here $\Delta t$ is chosen as 1 for simplicity, while it can be selected remarkably larger than 1 for the acceleration of dynamics simulations in practice.

where $\boldsymbol{m}_{t,ij}^{e,l} = \varphi_m \left( \boldsymbol{h}_{t,i}^{e,l}, \boldsymbol{h}_{t,j}^{e,l}, \|\vec{\boldsymbol{x}}_{t,i}^{e,l} - \vec{\boldsymbol{x}}_{t,j}^{e,l}\| \right)$, $\varphi_m$, $\varphi_x$ and $\varphi_h$ are Multi-Layer Perceptrons (MLPs), $\mathcal{N}(i)$ represents all neighboring nodes of the $i$-th node. Particularly, $\boldsymbol{m}_{t,ij}^{e,l}$ is an E(3)-invariant message from node $j$ to $i$, which can be used to aggregate and update $\boldsymbol{h}_{t,i}^{e,l}$ features via $\varphi_h$; as for the update of $\vec{\boldsymbol{x}}_{t,i}^{e,l}$, $\varphi_x$ is used to compute a 1D scalar $\varphi_x(\boldsymbol{m}_{t,ij}^{e,l})$ which is then left-multiplied with $\vec{\boldsymbol{x}}_{t,i}^{e,l} - \vec{\boldsymbol{x}}_{t,j}^{e,l}$ to preserve directional information.

The temporal module employs an equivariant self-attention mechanism to model inter-frame dependencies and dynamical patterns for each node, with shared parameters across all nodes. A key advantage of this attention strategy is its ability to naturally process variable-length inputs, aligning perfectly with our objectives. Unlike the full-attention mechanism employed in traditional Transformer's encoder, we adopt a causal attention paradigm to maintain temporal consistency, a choice empirically validated in ablation studies. Further details on this causal attention paradigm are provided in § B.5. The temporal message passing can be formally characterized by first computing the query, key, and value features: $\boldsymbol{q}_{t,i}^{e,l+1} = \varphi_{qe}(\boldsymbol{h}_{t,i}^{e,l+1})$, $\boldsymbol{k}_{s,i}^{e,l+1} = \varphi_{ke}(\boldsymbol{h}_{s,i}^{e,l+1})$, $\boldsymbol{v}_{s,i}^{e,l+1} = \varphi_{ve}(\boldsymbol{h}_{s,i}^{e,l+1})$, which generate the attention weights via $\alpha_{ts,i}^{e,l+1} = \texttt{Softmax}(\langle \boldsymbol{q}_{t,i}^{e,l+1}, \boldsymbol{k}_{s,i}^{e,l+1} \rangle)$. Here $\varphi_{qe}, \varphi_{ke}, \varphi_{ve}$ denote the query/key/value projection MLPs, and $\alpha_{ts,i}^{e,l+1}$ quantifies the temporal correlation between frame $t$ and $s$. With the derived attentions, we update the features by:

$$
\begin{aligned}
\boldsymbol{h}_{t,i}^{e,l+2} &= \boldsymbol{h}_{t,i}^{e,l+1} + \varphi_{ht} \left( \boldsymbol{h}_{t,i}^{e,l+1}, \sum_{s=1}^{t} \alpha_{ts,i}^{e,l+1} \boldsymbol{v}_{s,i}^{e,l+1} \right), \\
\vec{\boldsymbol{x}}_{t,i}^{e,l+2} &= \vec{\boldsymbol{x}}_{t,i}^{e,l+1} + \sum_{s=1}^{t} \alpha_{ts,i}^{e,l+1} \varphi_{xt} \left( \boldsymbol{v}_{s,i}^{e,l+1} \right) \cdot \left( \vec{\boldsymbol{x}}_{t,i}^{e,l+1} - \vec{\boldsymbol{x}}_{s,i}^{e,l+1} \right).
\end{aligned}
\tag{2}
$$

Consequently, the encoder yields a refined and compressed representation of the sequential features, encapsulating both spatial and temporal dependencies in a more compact and informative format.

### (2) Equivariant Decoder

Similar to the encoder, our decoder also applies $L$ equivariant spatiotemporal blocks. The main difference is that the decoder processes not only the spatiotemporal graph but also the TDG $\mathcal{G}_\delta$. Below we first define the concept of the TDG in prior to the introduction of the decoder architecture.

**Temporal Difference Graph $\mathcal{G}_\delta$.** Existing methods tend to amplify cumulative errors as the rollout distance increases. We circumvent this issue by conditioning each prediction exclusively on the integration of the last predicted frame and $\mathcal{G}_\delta$, significantly reducing error propagation during multi-step autoregressive prediction. This approach is motivated by the strong local correlation observed between adjacent frames in physical dynamics. Predicting the difference between the current frame and the next one may be easier than directly predicting the next frame (see the theoretical explanations in § B.8). To do so, we first initialize TDG as $\mathcal{G}_\delta = \mathcal{G}_{T+r-1} - \mathcal{G}_{T+r-2}$, namely, for each node $i$: $\boldsymbol{h}_{\delta,i} = \boldsymbol{h}_{T+r-1,i} - \boldsymbol{h}_{T+r-2,i}$, $\vec{\boldsymbol{x}}_{\delta,i} = \vec{\boldsymbol{x}}_{T+r-1,i} - \vec{\boldsymbol{x}}_{T+r-2,i}$. After the initialization, $\mathcal{G}_\delta$ along with $\mathcal{G}_{1:T+r-1}$ will be fed into the decoder. Within the decoder, $\mathcal{G}_\delta$ is considered as a global graph which interacts with each other graph to gather the global information to refine its node features layer by layer. Although our final prediction is made by adding the last frame to the delta graph, the process of learning the delta graph itself involved interactions with all previous frames. This ensures full historical context influences and contributes the predictions, as evidenced by visualized attention map in Fig. 5. The second-order extension of TDG is shown in § B.8, and comparisons with other methods are presented in § B.9. TDG Diagram is shown in Fig. 6(a). § B.7 and § B.8.1 detail why predicting frame differences is simpler than direct frame prediction and the superiority of the TDG implementation, respectively.

We now present the formulation of the spatiotemporal block. For conciseness, we denote the $t$-th frame (including the TDG frame) of the $l$-th layer as $\mathcal{G}_t^{d,l}(\boldsymbol{h}_t^{d,l}, \vec{\boldsymbol{X}}_t^{d,l})$. The spatial module follows the same mechanism as descried in Eq. (1) for all frames, including the TDG. In contrast, the temporal module performs causal attention (Eq. (2)) among all frames, excluding the TDG. In particular, the update of the TDG in the temporal module is given by:

$$h_{\delta,i}^{d,l+2} = h_{\delta,i}^{d,l+1} + \varphi_{ht}\left(h_{\delta,i}^{d,l+1}, \sum_{s=1}^{T+r-1} \alpha_{\delta s,i}^{d,l+1} v_{s,i}^{d,l+1}\right),$$

$$\vec{x}_{\delta,i}^{d,l+2} = \vec{x}_{\delta,i}^{d,l+1} + \sum_{s=1}^{T+r-1} \alpha_{\delta s,i}^{d,l+1} \varphi_{xt}\left(v_{s,i}^{d,l+1}\right) \cdot \left(\vec{x}_{\delta,i}^{d,l+1} - (\vec{x}_{s,i}^{d,l+1} - \bar{x})\right), \quad (3)$$

where $\alpha_{\delta s,i}^{d,l+1}$ stands for the attention weight between $\mathcal{G}_\delta$ and the $s$-th frame. Importantly, since $\vec{x}_{\delta,i}^{d,l+2}$ should be translation invariant, we have carried out mean reduction $\vec{x}_{s,i}^{d,l+1} - \bar{x}$ beforehand for the update of $\vec{x}_{\delta,i}^{d,l+2}$, where $\bar{x}$ is the mean of all nodes across all frames in $\mathcal{G}_{1:T}$. The mean reduction term $\bar{x}$ is computed only once based on the initial input trajectory and is not dynamically recomputed at each step of the autoregressive rollout process.

Finally, we employ a cross-attention mechanism, utilizing $(\mathcal{G}_{1:T+r-1}^{d,l+2}, \mathcal{G}_\delta^{d,l+2})$ as query to extract useful information from encoder-compressed sequence $\mathcal{G}_{1:T+r-1}^{e,l+2}$. This formulation enables the model to dynamically focus on relevant historical information, thereby enhancing the fidelity and contextual relevance of spatiotemporal representations. This mechanism potentially leads to improved model performance and generalization capabilities in capturing complex spatiotemporal dynamics. The update procedure computes the query, key, and value features through dedicated MLP projections: $q_{t,i}^{d,l+2} = \varphi_{qd}(h_{t,i}^{d,l+2})$, $k_{s,i}^{e,l+2} = \varphi_{kd}(h_{s,i}^{e,l+2})$, $v_{s,i}^{e,l+2} = \varphi_{vd}(h_{s,i}^{e,l+2})$, which leads to the attention weights $\alpha_{ts,i}^{d,l+2} = \texttt{Softmax}\left(\langle q_{t,i}^{d,l+2}, k_{t,i}^{e,l+2}\rangle\right)$. Then we update the $h_{t,i}^{d,l+2}$ and $\vec{x}_{t,i}^{d,l+2}$ as follows:

$$h_{t,i}^{d,l+3} = h_{t,i}^{d,l+2} + \varphi_{ht}\left(h_{t,i}^{d,l+2}, \sum_{s=1}^{t} \alpha_{ts,i}^{d,l+2} v_{s,i}^{e,l+2}\right),$$

$$\vec{x}_{t,i}^{d,l+3} = \vec{x}_{t,i}^{d,l+2} + \sum_{s=1}^{t} \alpha_{ts,i}^{d,l+2} \varphi_{xt}\left(v_{s,i}^{e,l+2}\right) \cdot \left(\vec{x}_{t,i}^{d,l+2} - \vec{x}_{s,i}^{e,l+2}\right). \quad (4)$$

Note that when performing the cross-attention between $\mathcal{G}_\delta^{d,l+2}$ and $\mathcal{G}_s^{e,l+2}$, we will first accomplish mean reduction for the coordinates in $\mathcal{G}_s^{e,l+2}$. A brief introduction to aforementioned equations, along with an overview of the overall process of how node/edge features evolve, is provided in the § B.1.

**(3) Autoregressive Mean Squared Error Loss**

We denote the output of the decoder as $\vec{x}_{T+r-1,i}'$ for the last frame and $\vec{x}_{\delta,i}'$ for the TDG. The prediction of the next frame is given by $\vec{x}_{T+r,i}' = \vec{x}_{T+r-1,i}' + \vec{x}_{\delta,i}'$. This predicted frame is then concatenated into the original sequence, forming a new input for the model, and this process continues until we predict the final frame $\vec{x}_{T+R}'$.

We collect all the autoregressive predictions and employ Mean Squared Error (MSE) to compute the prediction errors, which can be formalized as: $\mathcal{L} = \sum_{r=1}^{R} \sum_{i=1}^{N} \texttt{MSE}(\vec{x}_{T+r,i}, \vec{x}_{T+r,i}')$. To mitigate cumulative errors during the autoregressive inference, we implement a teacher-forcing strategy in training phase, implying that $\vec{x}_{T+r,i}'$ is estimated through the input of ground-truth sequence rather than predicted one. It is noteworthy that all baselines adhere to same strategies during both training and testing, ensuring a fair analysis. An in-depth comparison between our EST and Video Transformer frameworks is presented in § B.2. A fundamental characteristic of our EST is its E(3)-equivariance property, and the proof is provided in § A. Furthermore, our framework is extensible to various symmetry groups beyond E(3), and the proof can be found in § B.4.

## 4 EXPERIMENTS

In this section, we evaluate the performance of EST on autoregressive prediction tasks across datasets of varying scales, encompassing molecules (§ 4.1), human motions (§ 4.2), and proteins (§ 4.3). The rationale for adopting these three datasets is delineated in § C.1. To accelerate the dynamics simulations, we follow the sampling approach utilized in previous works (Huang et al., 2022) to acquire the subset of the trajectories for training, validation and testing. Specifically, we randomly

Table 1: Predicted MSE ($\times 10^{-2}$) on MD17 dataset with 10 rollout steps.

| | Aspirin | Benzene | Ethanol | Malonaldehyde | Naphthalene | Salicylic | Toluene | Uracil |
|---|---|---|---|---|---|---|---|---|
| ST_GNN | $9.403_{\pm 0.150}$ | $1.942_{\pm 0.086}$ | $2.650_{\pm 0.001}$ | $7.203_{\pm 0.102}$ | $4.311_{\pm 0.172}$ | $5.565_{\pm 0.251}$ | $4.530_{\pm 0.061}$ | $4.028_{\pm 0.374}$ |
| ST_TFN | $7.974_{\pm 0.025}$ | $2.084_{\pm 0.001}$ | $2.441_{\pm 0.001}$ | $6.228_{\pm 0.066}$ | $4.768_{\pm 0.078}$ | $6.737_{\pm 0.024}$ | $4.041_{\pm 0.198}$ | $5.672_{\pm 0.098}$ |
| STGCN | $8.079_{\pm 0.001}$ | $1.993_{\pm 0.004}$ | $2.786_{\pm 0.001}$ | $6.464_{\pm 0.001}$ | $5.829_{\pm 0.001}$ | $6.739_{\pm 0.001}$ | $4.724_{\pm 0.001}$ | $6.119_{\pm 0.001}$ |
| ST_SE(3)-Tr. | $6.943_{\pm 0.082}$ | $2.085_{\pm 0.006}$ | $2.079_{\pm 0.001}$ | $5.775_{\pm 0.016}$ | $4.443_{\pm 0.046}$ | $5.577_{\pm 0.021}$ | $3.292_{\pm 0.004}$ | $4.914_{\pm 0.042}$ |
| ST_EGNN | $7.945_{\pm 0.040}$ | $3.764_{\pm 1.834}$ | $1.385_{\pm 0.001}$ | $4.661_{\pm 0.084}$ | $4.226_{\pm 0.752}$ | $6.214_{\pm 0.232}$ | $3.405_{\pm 0.178}$ | $3.303_{\pm 0.291}$ |
| AGL-STAN | $11.885_{\pm 0.697}$ | $5.813_{\pm 0.278}$ | $3.052_{\pm 0.286}$ | $21.715_{\pm 0.617}$ | $2.248_{\pm 0.198}$ | $4.871_{\pm 0.928}$ | $1.909_{\pm 0.053}$ | $1.697_{\pm 0.329}$ |
| EqMotion | $8.433_{\pm 0.001}$ | $4.724_{\pm 0.001}$ | $4.275_{\pm 0.001}$ | $6.787_{\pm 0.001}$ | $6.538_{\pm 0.001}$ | $7.227_{\pm 0.001}$ | $4.922_{\pm 0.001}$ | $6.369_{\pm 0.001}$ |
| ESTAG | $2.553_{\pm 0.414}$ | $1.524_{\pm 0.142}$ | $0.977_{\pm 0.001}$ | $2.758_{\pm 0.794}$ | $2.278_{\pm 0.211}$ | $2.239_{\pm 0.576}$ | $1.733_{\pm 0.591}$ | $1.600_{\pm 0.237}$ |
| EST-F | $\underline{2.345}_{\pm 0.077}$ | $\underline{0.873}_{\pm 0.098}$ | $\underline{0.968}_{\pm 0.001}$ | $\mathbf{1.442}_{\pm 0.025}$ | $\underline{1.297}_{\pm 0.185}$ | $\underline{1.895}_{\pm 0.034}$ | $\mathbf{0.957}_{\pm 0.117}$ | $\underline{1.470}_{\pm 0.234}$ |
| EST-V | $\mathbf{2.196}_{\pm 0.075}$ | $\mathbf{0.480}_{\pm 0.050}$ | $\mathbf{0.940}_{\pm 0.001}$ | $\underline{1.762}_{\pm 0.054}$ | $\mathbf{0.988}_{\pm 0.016}$ | $\mathbf{1.733}_{\pm 0.031}$ | $\underline{1.002}_{\pm 0.063}$ | $\mathbf{1.087}_{\pm 0.055}$ |

select a starting point and subsequently sample $T + 20$ timestamps. The initial $T$ timestamps serve as input observations for the models, while the subsequent 10, 15, and 20 timestamps are future frames to be predicted, depending on the specific task settings about the rollout steps. In § 4.4, we conduct ablation studies to explore the impact of each component on model performance. Additionally, we perform exploratory experiments to identify potentially optimal encoder-decoder structures. More detailed experimental results and analysis are also provided in § D and § E.

**Baselines and Metrics.** We benchmark our method against the following baseline approaches, including GNNs such as ST_GNN (Gilmer et al., 2017), STGCN (Yu et al., 2017), and AGL-STAN (Sun et al., 2022), as well as equivariant GNNs such as ST_TFN (Thomas et al., 2018), ST_SE(3)-Tr. (Fuchs et al., 2020), ST_EGNN (Satorras et al., 2021), ST_GMN (Huang et al., 2022), Eqmotion (Xu et al., 2023), and ESTAG (Wu et al., 2024). Models prefixed with "ST" indicate we have adapted them to accommodate multi-frame input by incorporating trivial spatio-temporal aggregation, following the setup in Wu et al. (2024). We also report the performance for two versions of the EST: EST-V and EST-F. EST-V (EST-Variable) receives variable-length historical frames as input, while EST-F (EST-Fixed) adopts the same configuration as other baselines, accepting only fixed-length historical frame inputs. We employ the sum of MSEs across all predicted frames during the rollout process as our evaluation metric.

## 4.1 MOLECULAR DYNAMICS

**Dataset and Implementation.** MD17 comprises dynamic trajectories generated by MD simulations for eight distinct small molecules (*e.g.*, aspirin, benzene, etc). We utilize atomic numbers as invariant input features $\boldsymbol{h}_{t,i}$. Further details, including hyper-parameters are provided in § C.2.

**Results.** Table 1 shows performance of all models on MD17 with a 10-step rollout. Key insights include: **1.** Our model achieves SOTA performance on all eight small molecules, highlighting its superiority in autoregressive prediction. **2.** EST-V outperforms EST-F on most molecules, indicating our architecture's ability to effectively utilize extended historical information for improved accuracy. **3.** Our method surpasses the previous SOTA method ESTAG on seven of the eight molecules significantly, with ethanol being the exception. We attribute this to ethanol's simple structure (only three heavy atoms), which may minimize performance differences between models. **4.** Additionally, § E.1 provides an intuitive comparison of prediction accuracy across models at each rollout step.

We also evaluate performance on MD17 using extended rollout steps of 15, 20 and 100, allowing us to assess longer-term predictions more effectively. Besides, we modify most baseline models to accept full historical inputs by employing the temporal self-attention mechanism, except for AGL-STAN and Eqmotion due to architectural constraints. Detailed results and corresponding analyses are presented in Tables 14 to 17 of § E. EST consistently outperforms state-of-the-art methods across almost all molecules, even under these more challenging conditions. Notably, the performance gap widens for several molecules as rollout steps increase, validating EST's capability to suppress error propagation and achieve high-fidelity autoregressive predictions. Additionally, we present the visualization results of the attention map for EST-V in § F. The complexity analysis of the model is provided in § E.8. A comparison with EST and Hamiltonian or Lagrangian neural networks (Greydanus et al., 2019;

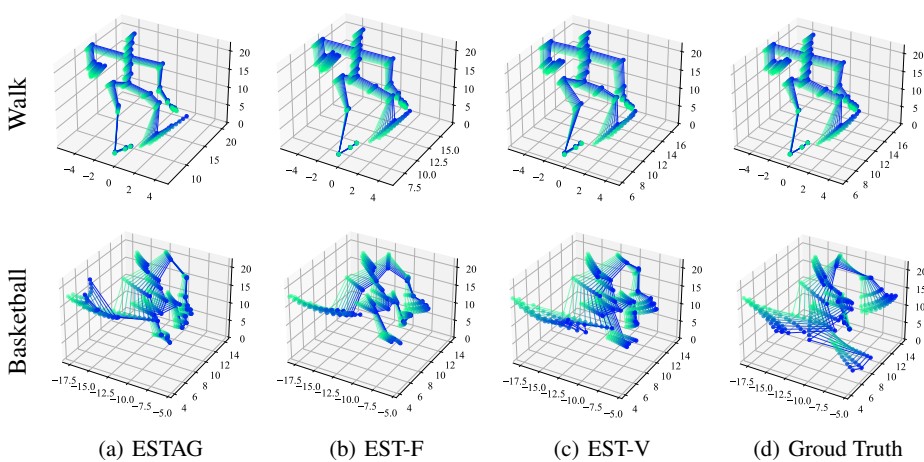

(a) ESTAG        (b) EST-F        (c) EST-V        (d) Groud Truth

Figure 3: Visualizations of predicted human motions: `Walk` (top) and `Basketball` (bottom).

Cranmer et al., 2020) is provided in § B.10. We assess the model's ability to generalize to continuous-time prediction, unseen rapid changes, and systems with varying components or interactions, as detailed in § E.5 to E.7.

## 4.2 MOTION CAPTURE

**Dataset and Implementation.** We evaluate our model's performance across various scenarios depicting 3D trajectories of human motion. We focus on two motions: Subject #35 (`Walk`) and Subject #102 (`Basketball`). The more details, including adaptive modifications to other baselines and hyper-parameter settings, are presented in § C.3.

**Results.** As illustrated in Table 2, our model achieves the best perofrmance across different autoregressive prediction tasks on the Motion Capture dataset. In this table, we denote predicted MSE values exceeding 1000 with a dash (-) symbol. Additionally, we omit the results for Eqmotion, as its predicted MSE values surpassed 1000 across all the settings. It is noteworthy that due to the complexity and rapid changes inherent in motion capture, the predicted MSE for all models increases rapidly as the rollout steps extends. Correspondingly, the performance gap between our method and other baselines widens as the number of rollout steps increases. Additionally, the predicted MSE results per step are provided in § D.1. Furthermore, Fig. 3 showcases the visualization of the predicted motions in `Basketball` and `Walk` by various methods.

## 4.3 PROTEIN DYNAMICS

**Dataset and Implementation.** We leveraged MDAnalysis toolkit (Gowers et al., 2019) to facilitate exploration of model's dynamics simulation capabilities on Adk equilibrium trajectory dataset (Seyler & Beckstein, 2017). To mitigate the computational burden associated with the large number of atoms in protein data, we represent each residue solely by its backbone atoms. The more details, including adaptive modifications to other baselines and hyper-parameter settings, are presented in § C.4.

**Results.** Table 3 illustrates the performance of all models on protein dataset. Our EST consistently achieves superior performance across various prediction tasks with rollout steps of 10, 15, and 20. This demonstrates that even in systems with complex structures such as proteins, EST can still efficiently extract and utilize critical information from historical timestamps, enabling more accurate autoregressive predictions. The prediction results for all-atom protein dynamics are shown in § E.9.

## 4.4 ABLATION STUDIES

To validate the contribution of each module to EST's overall performance and to explore the intricacies of Encoder-Decoder framework in physical dynamics scenarios, we conduct ablation and exploratory experiments. The results are presented in Table 4. Our observations are as follows: **1.** Disregarding

Table 2: Predicted MSE ($\times 10^{-1}$) on Motion dataset. We denote predicted MSE values exceeding 1000 with a dash (-) symbol.

Table 3: Predicted MSE ($\times 10^{-2}$) on Protein dataset.

| | Walk | | | Basketball | | |
|---|---|---|---|---|---|---|
| | R=10 | R=15 | R=20 | R=10 | R=15 | R=20 |
| ST_GNN | $18.560_{\pm 1.215}$ | $55.062_{\pm 8.120}$ | $97.593_{\pm 9.933}$ | - | - | - |
| ST_TFN | $19.689_{\pm 0.631}$ | $99.021_{\pm 28.136}$ | $201.075_{\pm 11.1}$ | $178.689_{\pm 2.477}$ | $593.498_{\pm 32.395}$ | - |
| ST_GCN | $1.870_{\pm 0.001}$ | $7.418_{\pm 0.001}$ | $13.899_{\pm 0.001}$ | $87.185_{\pm 0.001}$ | $312.096_{\pm 0.001}$ | $531.535_{\pm 0.001}$ |
| ST_SE(3)-Tr. | $8.196_{\pm 1.000}$ | $44.096_{\pm 7.873}$ | $164.483_{\pm 9.169}$ | $183.933_{\pm 6.797}$ | $580.178_{\pm 12.324}$ | - |
| ST_EGNN | $35.863_{\pm 3.156}$ | - | - | - | - | - |
| AGL-STAN | $42.409_{\pm 0.001}$ | $64.652_{\pm 0.001}$ | $170.848_{\pm 0.001}$ | - | - | - |
| ESTAG | $1.418_{\pm 0.087}$ | $5.907_{\pm 0.885}$ | $17.431_{\pm 3.424}$ | $10.209_{\pm 0.071}$ | $54.513_{\pm 1.682}$ | $175.950_{\pm 3.073}$ |
| EST-F | $\mathbf{0.931}_{\pm 0.097}$ | $\underline{4.192}_{\pm 0.765}$ | $\mathbf{11.439}_{\pm 2.408}$ | $\underline{9.712}_{\pm 0.203}$ | $\underline{49.754}_{\pm 0.587}$ | $\underline{155.296}_{\pm 9.479}$ |
| EST-V | $\underline{1.095}_{\pm 0.142}$ | $\mathbf{4.084}_{\pm 0.311}$ | $\underline{12.708}_{\pm 1.466}$ | $\mathbf{9.658}_{\pm 0.072}$ | $\mathbf{49.374}_{\pm 1.410}$ | $\mathbf{148.988}_{\pm 6.631}$ |

| Method | R=10 | R=15 | R=20 |
|---|---|---|---|
| ST_GNN | 2.196 | 3.108 | $\underline{4.202}$ |
| ST_GCN | 2.285 | 3.700 | 5.733 |
| ST_EGNN | $\underline{2.000}$ | $\underline{3.051}$ | 4.239 |
| AGL-STAN | 2.216 | 3.373 | 4.309 |
| ST_GMN | 2.006 | 3.056 | 4.247 |
| ESTAG | 2.009 | 3.065 | 4.259 |
| EST-F | 2.008 | 3.063 | 4.258 |
| EST-V | **1.911** | **2.971** | **4.048** |

Table 4: Ablation studies ($\times 10^{-3}$) on MD17 dataset with 10 rollout steps.

| | Aspirin | Benzene | Ethanol | Malonaldehyde | Naphthalene | Salicylic | Toluene | Uracil |
|---|---|---|---|---|---|---|---|---|
| w/o Equivariance | 30.299 | 2.660 | 21.896 | 24.474 | 35.048 | 31.365 | 32.874 | 31.252 |
| w/o TDG | 7.849 | 1.150 | 1.991 | 5.321 | 7.000 | 7.227 | 3.301 | 4.379 |
| Only TDG | 2.714 | 1.303 | 1.167 | 2.143 | 1.480 | 1.940 | 1.262 | 1.121 |
| Decoder-only | **1.680** | 0.545 | 1.104 | 2.071 | 1.261 | 1.929 | 1.420 | 1.327 |
| Full-Attention | 2.206 | 0.543 | 0.980 | 2.009 | **0.934** | 1.889 | 1.035 | 1.095 |
| Shared Parameters | 1.757 | 0.665 | 1.018 | 1.902 | 1.478 | **1.548** | 1.284 | 1.172 |
| EST-V | 2.195 | **0.480** | **0.940** | **1.761** | 0.988 | 1.733 | **1.002** | **1.087** |

equivariance led to a significant performance degradation across all molecules (Row 1). This underscores the critical role of physical symmetry in modeling 3D physical dynamics. **2.** To assess the impact of TDG, we remove TDG from decoder (Row 2). The results indicate the absence of TDG substantially impairs model performance, suggesting a strong correlation between physical trajectories and local frames of the input sequence, particularly the final two frames. **3.** We experiment with modifying the decoder input to solely include TDG (Row 3). This modification resultes in a notable performance decline, indicating the necessity of allowing TDG and original sequence to interact, thereby extracting crucial temporal information. **4.** To evaluate the benefits of the encoder-decoder structure, we implement a decoder-only structure (Row 4). The results demonstrate the encoder-decoder architecture outperforms the decoder-only variant for the majority of molecules. **5.** We adopt masking strategy from standard Transformer, replacing current causal-attention in encoder's self-attention module with full-attention (Row 5). This substitution led to a slight performance decrease for most molecules, suggesting that when performing self-attention on temporal data, the model must adhere to objective physical laws, allowing the current frame to be updated based solely on previous frames. **6.** To ascertain whether the performance improvement is due to the increased parameters introduced by the encoder-decoder architecture, we share parameters between encoder and decoder modules, effectively halving the total parameters of the model (Row 6). We observe performance decreases for some molecules, but improvements for others. Notably, when computing the average across all eight molecules, the shared-parameter version exhibited only a marginal difference in performance compared to the original version. This highlights the encoder-decoder framework's effectiveness in reducing error accumulation, leading to more stable predictions.

## 5 CONCLUSION

In this paper, we introduce Equivariant Spatiotemporal Transformers (EST), a novel framework integrating the Transformer architecture with physical symmetries to enable autoregressive simulation of physical dynamics. Leveraging the inherent characteristics of the Transformer, EST efficiently processes historical inputs of arbitrary length. The overall EST architecture ensures E(3)-equivariance, effectively encoding physical symmetries and enhancing the modeling capability for 3D objects. The incorporation of a Temporal Difference Graph (TDG) significantly mitigates cumulative errors generated during autoregressive process. Extensive experiments across various scales (molecules, proteins, and human motions) and diverse rollout steps demonstrate EST's superior performance in physical dynamics simulation. We envision EST serving as a robust baseline in this field, with broad applicability across various tasks, including drug discovery, robot control, and materials design.

## 6 ETHICS STATEMENT

This work complies with the ICLR Code of Ethics. This study did not involve any human subjects or animal experimentation. All datasets were obtained and used in accordance with their respective licensing and usage agreements. We confirm that no personally identifiable information was involved, and rigorous measures were taken to avoid any biases in the research process. The study design raises no ethical, privacy, or security concerns. We affirm our commitment to upholding the highest standards of transparency and integrity in our research.

## 7 REPRODUCIBILITY STATEMENT

We have made every effort to ensure that the results presented in this paper are reproducible. All code and datasets will be made publicly available upon acceptance of the paper to facilitate replication and verification. The experimental setup, including training epochs, model configurations, and hardware details, is described in detail in § C. We have also provided a full description of our method in § 3, to assist others in reproducing our experiments. Given the public availability of all datasets, we believe these steps will empower the research community to verify our findings and build upon them.

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

CONTENTS OF APPENDIX

## A  EQUIVARIANCE OF EST

We know that the E(3) symmetry can be decomposed into the symmetry on the three-dimensional translation group T(3) and the symmetry on the three-dimensional orthogonal group O(3). We will prove these two separately below.

### A.1  TRANSLATION EQUIVARIANCE OF EST

With mean reduction (Puny et al., 2021), translational variability can always be easily achieved. We just need to prove that subsequent operations are translation invariant. In fact, there are only Eqs. (1) to (3) require to prove its translation invariant. And we find that, all sums of the coefficients of these coordinate terms is $1 + (-1) = 0$, which indicates its translation invariance.

### A.2  O(3)-EQUIVARIANCE OF EST

O(3)-**equivariance of whole model.** Since the equivariance can be understood that group actions can be exchanged with mappings of any layer, so the equivariance of the entire model can be proved by proving that the individual modules of the model are equivariant. In general, we only need to prove the following three points:

1. O(3)-equivariance of spatial module.

2. O(3)-equivariance of temporal module.

3. O(3)-invariance of objective function.

Here we give the proof of O(3)-equivariance by the symbology from e3nn (Geiger & Smidt, 2022). O(3) consists of rotation and inverse, implying $O(3) = SO(3) \times C_i$, where SO(3) is the rotation group and $C_i = \{\mathfrak{e}, \mathfrak{i}\}$ denotes the inverse group. We thus specify the group representation of O(3) as

$$\rho^{(l)}(\mathfrak{rm}) := \sigma^{(l,p)}(\mathfrak{m})\boldsymbol{D}^{(l)}(\mathfrak{r}), \tag{5}$$

where $p \in \{\pm 1\}$ called *parity*. $\sigma^{(l)}(\mathfrak{m}) = 1$ for $\mathfrak{m} = \mathfrak{e}$ (the identity) , $\sigma^{(l)}(\mathfrak{m}) = p^l$ if $\mathfrak{m} = \mathfrak{i}$ (the inverse), and $\boldsymbol{D}^{(l)}(\mathfrak{r})$ is the Wigner-D matrix of $l$-degree. With Eq. (5) of $l$ be the representation, features be the solution of such equivariant constraint equation is call $l$-degree steerable features. Moreover, we donate a $l$-degree vector as $l\circ$ if $p = -1$ (inverse equivariant), and $l\mathrm{e}$ if $p = 1$ (inverse invariant).

**Example A.1** (Types of common equivariant/invariant features). Considering that the highest degree of features in this article does not exceed 1, we only introduce four common cases: $0\mathrm{e}$, $0\mathrm{o}$, $1\mathrm{o}$, and $1\mathrm{e}$.

- Scalar (*e.g.* charges, distance and potential energy) is invariant to both rotation and inverse, thus denoted by $0\mathrm{e}$.

- Pseudo-scalar (*e.g.* triple product of three 3-dimensional vectors, magnetic flux and helicity) is invariant to rotation but equivariant to inverse, thus denoted by $0\mathrm{o}$.

- Vector (*e.g.* postion, velocity and acceleration) is equivariant to both rotation and inverse, thus denoted by $1\mathrm{o}$.

- Pseudo-vector (*e.g.* angular momentum, torque and magnetic field vector) is equivariant to rotation but invariant to inverse, thus denoted by $1\mathrm{e}$.

In the scope of consideration of this article, there are only two types of operations to change the degree of features: $(0\mathrm{e}) \cdot (1\mathrm{o}) \rightarrow (1\mathrm{o})$ and transforming $(1\mathrm{o})$ into $(0\mathrm{e})$ through norm $\|1\mathrm{o}\| \rightarrow 0\mathrm{e}$. From this perspective, proving the equivariance of a model requires only pointing out all the types of variables. And it is worth noting that all inputs inside the function are invariants, which is denoted as $[0\mathrm{e}, 0\mathrm{e}, \ldots, 0\mathrm{e}] \rightarrow 0\mathrm{e}$.

O(3)-**equivariance of spatial module.** To prove the spatial module is O(3)-equivariant, we only require to prove message $\boldsymbol{m}_{t,ij}^{e,l}$ and updated node feature $\boldsymbol{h}_{t,i}^{e,l+1}$ are $0\mathrm{e}$-features and updated node

positions $\vec{\boldsymbol{x}}_{t,i}^{e,l+1}$ are 1o-features. Based on the symbology, we can easily see this by annotating Eq. (1) follows:

$$\underbrace{\boldsymbol{m}_{t,ij}^{e,l}}_{\texttt{0e}} = \underbrace{\varphi_m\left(\boldsymbol{h}_{t,i}^{e,l},\,\boldsymbol{h}_{t,j}^{e,l},\,\left\|\vec{\boldsymbol{x}}_{t,i}^{e,l} - \vec{\boldsymbol{x}}_{t,j}^{e,l}\right\|\right)}_{[\texttt{0e, 0e, }\|\texttt{1o}\|]\to\texttt{0e}},$$

$$\underbrace{\boldsymbol{h}_{t,i}^{e,l+1}}_{\texttt{0e}} = \underbrace{\boldsymbol{h}_{t,i}^{e,l}}_{\texttt{0e}} + \underbrace{\varphi_h\left(\boldsymbol{h}_{t,i}^{e,l},\,\textstyle\sum_{j\in\mathcal{N}(i)}\boldsymbol{m}_{t,j}^{e,l}\right)}_{[\texttt{0e, 0e}]\to\texttt{0e}}, \tag{6}$$

$$\underbrace{\vec{\boldsymbol{x}}_{t,i}^{e,l+1}}_{\texttt{1o}} = \underbrace{\vec{\boldsymbol{x}}_{t,i}^{e,l}}_{\texttt{1o}} + \frac{1}{|\mathcal{N}(i)|}\sum_{j\in\mathcal{N}(i)}\underbrace{\varphi_x\left(\boldsymbol{m}_{t,ij}^{e,l}\right)\cdot\left(\vec{\boldsymbol{x}}_{t,i}^{e,l} - \vec{\boldsymbol{x}}_{t,j}^{e,l}\right)}_{\texttt{0e}\,\cdot\,\texttt{1o}\to\texttt{1o}},$$

$\mathrm{O}(3)$-**equivariance of temporal module.** In fact, like most equivariant Graph Transformer models except $\mathrm{SE}(3)$-Transformer, the attention mechanism obtains 0e-features as follows:

$$\left.\begin{array}{l} \underbrace{\boldsymbol{q}_{t,i}^{e,l+1}}_{\texttt{0e}} = \underbrace{\varphi_{qe}(\boldsymbol{h}_{t,i}^{e,l+1})}_{[\texttt{0e}]\to\texttt{0e}} \\[2ex] \underbrace{\boldsymbol{k}_{s,i}^{e,l+1}}_{\texttt{0e}} = \underbrace{\varphi_{ke}(\boldsymbol{h}_{s,i}^{e,l+1})}_{[\texttt{0e}]\to\texttt{0e}} \\[2ex] \underbrace{\boldsymbol{v}_{s,i}^{e,l+1}}_{\texttt{0e}} = \underbrace{\varphi_{ve}(\boldsymbol{h}_{s,i}^{e,l+1})}_{[\texttt{0e}]\to\texttt{0e}} \end{array}\right\} \implies \underbrace{\alpha_{ts,i}^{e,l+1}}_{\texttt{0e}} = \underbrace{\texttt{Softmax}\left(\left\langle\boldsymbol{q}_{t,i}^{e,l+1},\,\boldsymbol{k}_{s,i}^{e,l+1}\right\rangle\right)}_{[\texttt{0e, 0e}]\to\texttt{0e}}, \tag{7}$$

After that, we only need to prove that updated node features $\boldsymbol{h}_{t,i}^{e,l+1}$ are 0e-features and updated node positions $\vec{\boldsymbol{x}}_{t,i}^{e,l+1}$ are 1o-features, just as we did for the $\mathrm{O}(3)$-equivariance of spatial module. The details are as follows:

$$\underbrace{\boldsymbol{h}_{t,i}^{e,l+2}}_{\texttt{0e}} = \underbrace{\boldsymbol{h}_{t,i}^{e,l+1}}_{\texttt{0e}} + \underbrace{\varphi_{ht}\left(\boldsymbol{h}_{t,i}^{e,l+1},\,\textstyle\sum_{s=1}^{t}\alpha_{ts,i}^{e,l+1}\boldsymbol{v}_{s,i}^{e,l+1}\right)}_{[\texttt{0e,0e}]\to\texttt{0e}},$$

$$\underbrace{\vec{\boldsymbol{x}}_{t,i}^{e,l+2}}_{\texttt{1o}} = \underbrace{\vec{\boldsymbol{x}}_{t,i}^{e,l+1}}_{\texttt{1o}} + \sum_{s=1}^{t}\underbrace{\alpha_{ts,i}^{e,l+1}\varphi_{xt}\left(\boldsymbol{v}_{s,i}^{e,l+1}\right)\cdot\left(\vec{\boldsymbol{x}}_{t,i}^{e,l+1} - \vec{\boldsymbol{x}}_{s,i}^{e,l+1}\right)}_{\texttt{0e}\cdot\texttt{0e}\cdot\texttt{1o}\to\texttt{1o}}. \tag{8}$$

Note that the formula form may be slightly different when initialized, but we only need to discuss the difference about translation equivariance/invariance, and we will not repeat the proof of rotation/inverse equivariance/invariance here.

$\mathrm{O}(3)$-**invariance of objective function.** Note that $\texttt{MSE}(\cdot)$ is a simple modulo length operator, so its $\mathrm{O}(3)$-invariance is obvious.

$$\underbrace{\mathcal{L}}_{\texttt{0e}} = \sum_{r=1}^{R}\sum_{i=1}^{N}\texttt{MSE}(\vec{\boldsymbol{x}}_{T+r,i}^{*},\,\vec{\boldsymbol{x}}_{T+r,i}) = \sum_{r=1}^{R}\sum_{i=1}^{N}\underbrace{\|\vec{\boldsymbol{x}}_{T+r,i}^{*} - \vec{\boldsymbol{x}}_{T+r,i}\|}_{\|\texttt{1o}\|\to\texttt{0e}}, \tag{9}$$

# B  ABOUT EST FRAMEWORK

## B.1  HOW NODE AND EDGE FEATURES EVOLVE

We introduce this process in detail in § 3.2, divided into the Equivariant Encoder and Equivariant Decoder modules. In the Equivariant Encoder, Equation 2 represents the update process of node/edge features on the graph corresponding to each timestep (temporal graph) using EGNN in the spatial dimension. Equation 3 represents the update of these features in the temporal dimension using the equivariant self-attention mechanism. In the Equivariant Decoder, we update the node/edge features of every temporal graph (excluding $\mathcal{G}_\delta$) in the input data using the same methods as Equation 2 and

Equation 3, sequentially in spatial and temporal dimensions. We omitted the repetitive description of this step in the original paper for brevity. Furthermore, due to the specificity of the TDG, we describe the update of node/edge features on the TDG in the temporal dimension separately using Equation 4. Finally, Equation 5 represents the process where we generate Queries from all temporal graphs updated by the Equivariant Decoder, and Keys and Values from those updated by the Equivariant Encoder, applying the equivariant cross-attention mechanism to update all node/edge features in the Equivariant Decoder.

## B.2 Comparison between Our EST and Video Transformer frameworks

Our work makes two key contributions. First, our research pioneers the adaptation of the conventional Transformer architecture for applications in the realm of physical dynamics, marking a significant progression in this field. Second, we underscore the non-trivial nature of this adaptation, as spatiotemporal tasks in video processing and 3D geometric graph processing differ fundamentally:

- **Data format**: While video tasks operate on sequential image data, our framework processes 3D structural data with inherent topological relationships after feature extraction.
- **Equivariance Considerations**: Unlike video-based models such as TimeSformer (Bertasius et al., 2021), which disregard equivariance, our approach explicitly incorporates this critical property. Rather than merely combining spatial EGNN modules with temporal attention modules, we introduce a novel design that ensures both intra-frame atomic interaction equivariance and temporal consistency. Central to this is the Temporal Difference Graph (TDG), which combines with attention mechanisms in the encoder and decoder while preserving equivariance through data decentralization. This design mitigates error accumulation effectively, further enhancing autoregressive prediction accuracy.

## B.3 Why We Choose EGNN

To the best of our knowledge, EGNN has been extensively validated for its equivariance properties and has garnered significant attention in the research community, with numerous studies published in top-tier conferences and journals (Hoogeboom et al., 2022; Xu et al., 2024; Wang et al., 2024b;a). Our decision to adopt EGNN as the backbone is motivated by the following considerations:

- Extending new architectures based on EGNN is a well-established practice, as evidenced by recent advancements such as EDM (Hoogeboom et al., 2022) and EGNO (Xu et al., 2024).
- Although more advanced equivariant graph neural networks may yield superior performance, such an approach would introduce ambiguity in attributing performance gains to either the equivariant framework or our proposed EST network.
- Architectures relying on high-degree steerable features, such as TFN (Thomas et al., 2018) and Equiformer (Liao & Smidt, 2022), impose substantial computational overhead, rendering them impractical due to their prohibitive time complexity.

Additionally, we have conducted experiments by replacing EGNN with TFN as our backbone, and the experimental results are shown in the second row of Table 5. The results demonstrate a certain performance improvement compared to the original ST_TFN, yet the performance still falls short of our proposed EST. This validates the effectiveness of our proposed method and indicates that, on the MD17 dataset, EGNN is a superior backbone choice compared to TFN.

Table 5: Performance on MD17 when using TFN as the backbone.

| Method | Aspirin | Benzene | Ethanol | Malonaldehyde | Naphthalene | Salicylic | Toluene | Uracil |
|--------|---------|---------|---------|---------------|-------------|-----------|---------|--------|
| ST_TFN | 7.974 | 2.084 | 2.441 | 6.228 | 4.768 | 6.737 | 4.041 | 5.672 |
| EST w/ TFN | 7.782 | 1.181 | 2.309 | 4.867 | 3.529 | 5.891 | 1.966 | 4.791 |
| EST | **2.196** | **0.480** | **0.940** | **1.762** | **0.988** | **1.733** | **1.002** | **1.087** |

We further add the comparison with advancing higher-order equivariant architecture such as Equiformer. Notably, the computational cost of Equiformer is substantial due to its use of higher-order

features, particularly under our rollout configuration. Given the prohibitive training time required for the case of rollout=10, we adopt an efficient alternative by predicting 10 frames simultaneously, instead of the autoregressive prediction. The results are documented in Table 6. While Equiformer exhibits modest performance gains over other baselines on several molecules, its performance still lags behind our EST approach by a large margin.

Table 6: Performance between EST and Equiformer on MD17.

| Method | Aspirin | Benzene | Ethanol | Malonaldehyde | Naphthalene | Salicylic | Toluene | Uracil |
|---|---|---|---|---|---|---|---|---|
| Equiformer | 10.13 | 2.00 | 1.88 | 8.05 | 3.43 | 5.79 | 2.09 | 4.38 |
| EST | **2.196** | **0.480** | **0.940** | **1.762** | **0.988** | **1.733** | **1.002** | **1.087** |

## B.4    APPLIED TO DIFFERENT DYNAMIC SYSTEMS

Our proposed TDG is designed with a high degree of adaptability, facilitating its customization for handling both SE(3)-equivariance and O(3)-equivariance. The latter accounts for partial symmetries under the influence of external forces, such as gravitational effects. Achieving these equivariances can be realized through methodologies inspired by those detailed in DiffSBDD (Schneuing et al., 2024) and SGNN (Han et al., 2022a), respectively. Essential modifications are made to the coordinate update formula specified in Eq. (1). We first represent the centroid of the geometric graph as $h_{t,c}^{e,l}$, then the Eq. (1) can be updated to handle SE(3)-equivariance as follows:

$$
\begin{aligned}
\vec{x}_{t,i}^{e,l+1} = \vec{x}_{t,i}^{e,l} + \frac{1}{|\mathcal{N}(i)|} \sum_{j \in \mathcal{N}(i)} \Big[ & \varphi_x(m_{t,ij}^{e,l}) \cdot (\vec{x}_{t,i}^{e,l} - \vec{x}_{t,j}^{e,l}) \\
& + \varphi_\times(m_{t,ij}^{e,l}) \cdot \left( (\vec{x}_{t,i}^{e,l} - \vec{x}_{t,c}^{e,l}) \times (\vec{x}_{t,j}^{e,l} - \vec{x}_{t,c}^{e,l}) \right) \Big].
\end{aligned}
\tag{10}
$$

Moreover, the Eq. (1) can be updated to handle O(3)-equivariance as follows:

$$
\begin{aligned}
\vec{x}_{t,i}^{e,l+1} = \vec{x}_{t,i}^{e,l} + \frac{1}{|\mathcal{N}(i)|} \sum_{j \in \mathcal{N}(i)} \Big[ & \varphi_x\left(m_{t,ij}^{e,l}, \vec{g}^\top(\vec{x}_{t,i}^{e,l} - \vec{x}_{t,j}^{e,l})\right) \cdot \left(\vec{x}_{t,i}^{e,l} - \vec{x}_{t,j}^{e,l}\right) \\
& + \varphi_g\left(m_{t,ij}^{e,l}, \vec{g}^\top(\vec{x}_{t,i}^{e,l} - \vec{x}_{t,j}^{e,l})\right) \cdot \vec{g} \Big].
\end{aligned}
\tag{11}
$$

## B.5    CAUSAL ATTENTION PARADIGM

We integrate the causal attention mechanism into temporal module of EST. This design is motivated by the inherent causality in time series data, where predictions for the current frame must rely exclusively on historical frames to preserve physical consistency. To enforce this constraint, a masking mechanism is implemented, restricting the attention of the current frame to preceding frames while excluding future frames. This stands in contrast to conventional self-attention approaches, which permit unrestricted attention across all positions in the sequence, regardless of temporal order. The efficacy of such causal attention strategies has been extensively validated in prior work, particularly in the domain of natural language processing (Vaswani, 2017; Touvron et al., 2023).

## B.6    THE SELECTION OF EDGE FEATURES

For edge features, we actually experimented with incorporating edge features (bond types) but found that they provided little improvement in performance. To reduce model complexity, we chose not to include additional edge features in our experiments. Regarding angular features, as mentioned in Section 3.3 of PaiNN (Schütt et al., 2021), if distance information is introduced at every layer, the stacked layers can implicitly model angular information within the network. In other words, our model can learn angular geometries in fact. It is worth mentioning that our model is highly flexible and can incorporate various edge features and angular features.

### B.7 Why Is Predicting the Difference from the Previous Frame Simpler Than Predicting the Next Frame Directly?

We explain this from three aspects. First, in physical dynamics, time steps are typically short, leading to strong local correlations between adjacent frames. This means $\vec{X}_{T+1}$ is highly similar to $\vec{X}_T$. If the model is tasked with directly predicting $\vec{X}_{T+1}$, it must first learn to "copy" the majority of information from $\vec{X}_T$ (i.e., learn an identity mapping) and then learn the minute change $\Delta\vec{X}$. Learning a perfect identity mapping is difficult and inefficient for deep networks. Conversely, if the model is tasked with predicting the difference $\Delta\vec{X}$ (as done by the TDG in EST), the task becomes much simpler. $\vec{X}_T$ is preserved as a shortcut via the final addition $\vec{X}_{T+1} = \vec{X}_T + \Delta\vec{X}$. Second, the difference $\Delta\vec{X}$ is typically a sparser signal closer to zero compared to the absolute coordinates $\vec{X}_{T+1}$. For neural networks, learning to predict small fluctuations near zero is generally easier than predicting large absolute values potentially far from the origin. Finally, this hypothesis is strongly supported by our ablation studies. In Table 4, we show the results for "w/o TDG". This variant leads to a significant performance drop. This demonstrates that weakening this explicit differential/dynamic modeling mechanism severely impairs the model's ability to capture physical trajectories.

### B.8 Analysis of TDG

Our implementation of TDG was not simply about predicting differences; instead, we processed the delta graph as an additional temporal frame, initialized from the last two frames' difference, and it conducted specific updates within the model. Specifically, in the decoder, the delta graph interacts with all historical frames via self/cross-attention. Furthermore, to ensure that the delta graph remains translation-invariant, we subtracted the mean coordinates from each frame during the attention process, as detailed in Eq. (3). In the following, we provide detailed analysis, theoretical justification, and experiment results to verify the effectiveness of TDG.

#### B.8.1 Why the Specific Implementation of the TDG Is Superior

The core reason for designing the TDG as an independent interaction graph token is the need to capture "global dynamics evolution trends" rather than merely calculating "local instantaneous velocity". Traditional simple velocity prediction schemes (such as GNS) typically rely on only the most recent frames to calculate current acceleration or velocity, which is a local operation. In contrast, although the TDG is initialized as the difference between the last two frames, upon entering the decoder, it interacts with all historical frames via the attention mechanism. This implies that TDG serves not merely as a predictive target, but also as a dynamic learner. It aggregates long-range dependency information from the entire historical trajectory (e.g., the oscillation cycle of a molecule or the long-term momentum trend of a macroscopic object) to dynamically refine the current update.

To demonstrate that this interactive TDG is superior to simple velocity prediction, we directly compared it with the GNS model, which employs simple velocity/acceleration prediction strategies, in § B.9. The experimental results, shown in Table 8, indicate that the error of EST on the MD17 dataset is an order of magnitude lower than that of GNS (e.g., Aspirin: 2.196 vs 23.545). This substantial performance gap proves that merely predicting velocity is insufficient; refining this difference prediction through TDG's global interaction mechanism is the key factor.

#### B.8.2 Theoretical Justification

Instead of directly predicting the coordinates at time $T+1$, we first predict the velocity at time $T$ (i.e., $\vec{X}'_\delta$) and then compute the coordinates $\vec{X}_{T+1}$. This approach can be justified from the perspective of Taylor expansion (Linnainmaa, 1976):

$$\vec{X}_{t+\Delta t} = \vec{X}_t + \vec{V}_t \cdot \Delta t + O(\Delta t^2). \tag{12}$$

By structuring our model in this way, we ensure that the error remains within $O(\Delta t^2)$. Besides, predicting velocity first and then integrating it to obtain the trajectory effectively incorporates prior physical knowledge, leading to better model convergence and more stable trajectory predictions.

### B.8.3 EXPERIMENTAL VALIDATION

The TDG technique was serendipitously identified during our experimental investigations. Preliminary trials revealed its consistent efficacy in enhancing performance across diverse datasets. Additionally, we examined the integration of higher-order differential terms, such as second-order differences, but empirical outcomes fell short of anticipated improvements. The detailed results are presented in Table 7.

From a theoretical perspective, we posit that this phenomenon may be conceptually aligned with first-order numerical techniques employed in solving differential equations. Notably, numerous AI methodologies are inspired by such approaches. For instance, the Euler-Maruyama method, widely utilized in recent Diffusion models for solving SDEs, exemplifies a first-order method.

Table 7: Performance of EST when using second-order differences on MD17.

| Method | Aspirin | Benzene | Ethanol | Malonaldehyde | Naphthalene | Salicylic | Toluene | Uracil |
|---|---|---|---|---|---|---|---|---|
| EST w/ second-order differences | 1.756 | 0.334 | 2.740 | 2.332 | 1.093 | 1.647 | 1.193 | 1.408 |
| EST | 2.196 | 0.480 | 0.940 | 1.762 | 0.988 | 1.733 | 1.002 | 1.087 |

## B.9 COMPARISON WITH GNS AND MGN

Our approach diverges significantly from the methodologies employed in GNS (Sanchez-Gonzalez et al., 2020) and MGN (Pfaff et al., 2021). In GNS, positional differences are interpreted as velocity vectors, which are subsequently combined with initial features to form input representations. The framework operates by forecasting acceleration values, which are then integrated to derive velocities and positional updates iteratively. Furthermore, GNS does not incorporate the inherent physical symmetries of 3D space, potentially limiting its capacity for generalization. Conversely, MGN adopts a dual strategy: in certain configurations, it predicts positional differences directly, while in others, it merges these differences with original features to construct node inputs. They transforms absolute coordinates into relative edge features, which may reduce the model's expressiveness.

In comparison, our framework integrates differences information through a distinct approach by conceptualizing the Temporal Difference Graph (TDG) as a global graph that interacts with all temporal graphs. At each iteration, node representations within the TDG undergo propagation and refinement. Rather than predicting additional differential values, our method leverages the refined node features from the TDG, fused with the final temporal graph, to derive the target coordinates. Additionally, our architecture is explicitly constructed to adhere to E(3) symmetry (3D translation and rotation symmetry). Furthermore, the TDG is meticulously designed to preserve equivariance, and we formally prove its compliance with E(3)-equivariant properties.

In conclusion, our approach exhibits substantial distinctions from GNS and MGN. To validate its effectiveness, we performed comparative experiments utilizing GNS on the MD17 dataset. The results, as presented in Table 9, reveal that our method achieves markedly superior performance relative to GNS, underscoring its superiority.

Table 8: Performance between EST and GNS on MD17.

| Method | Aspirin | Benzene | Ethanol | Malonaldehyde | Naphthalene | Salicylic | Toluene | Uracil |
|---|---|---|---|---|---|---|---|---|
| GNS | 23.545±0.001 | 2.530±0.002 | 15.793±0.001 | 17.878±0.001 | 12.768±0.006 | 22.498±0.001 | 12.109±0.001 | 21.299±0.001 |
| EST | 2.196±0.075 | 0.480±0.050 | 0.940±0.001 | 1.762±0.054 | 0.988±0.016 | 1.733±0.031 | 1.002±0.063 | 1.087±0.055 |

## B.10 COMPARISON WITH HNN AND LNN

While Hamiltonian Neural Networks (HNN) (Greydanus et al., 2019) and Lagrangian Neural Networks (LNN) (Cranmer et al., 2020) theoretically guarantee energy conservation by learning the system's Hamiltonian or Lagrangian, fitting a complex potential energy surface in high-dimensional spaces (such as molecular and protein systems) proves relatively difficult. This strict mathematical form often leads to optimization challenges and sensitivity to noise, resulting in underfitting. In contrast, EST leverages $E(3)$ equivariance and the Transformer architecture; its advantage lies in

its strong expressivity and flexibility. The Transformer architecture captures extremely complex spatiotemporal dependencies, while $E(3)$ equivariance ensures generalization across arbitrary coordinate systems. We further reproduce the HNN model on the standard MD17 benchmark and compare it with EST, as shown in Table 9. The results demonstrate that EST achieves significantly lower prediction errors than HNN across all molecular tasks, with performance improvements ranging from 3 to 7 times. This strongly suggests that for real-world, complex molecular dynamics data involving high-order interactions, the rigid physical constraints of HNN limit fitting capability. EST, by combining physical symmetry with the expressive power of Transformers, simulates the complex trajectories of microscopic particles more precisely while maintaining physical consistency.

Table 9: Performance between HNN and EST on MD17.

| Method | Aspirin | Benzene | Ethanol | Malonaldehyde | Naphthalene | Salicylic | Toluene | Uracil |
|--------|---------|---------|---------|---------------|-------------|-----------|---------|--------|
| HNN | 8.378±0.057 | 1.537±0.060 | 3.292±0.001 | 6.228±0.009 | 7.412±0.052 | 7.942±0.129 | 5.786±0.051 | 7.003±0.168 |
| EST | 2.196±0.075 | 0.480±0.050 | 0.940±0.001 | 1.762±0.054 | 0.988±0.016 | 1.733±0.031 | 1.002±0.063 | 1.087±0.055 |

### B.11 EXTENSION TO GAUGE EQUIVARIANCE

For the dynamical systems discussed in this paper, E(n)-equivariance suffices. However, for physical systems defined on more complex manifolds, gauge equivariance becomes necessary. Currently, our model cannot be directly applied to such cases because the backbone architecture lacks built-in gauge-equivariant properties. Nevertheless, if a gauge-equivariant backbone model is employed, our proposed delta graph construction and encoder-decoder framework can be readily adapted. Alternatively, discretizing the manifold into geometric graphs could enable effective application of our method. We are keen to explore these extensions in future work and evaluate the model's performance in such settings.

## C DATASET DETAILS

### C.1 THE REASON OF SELECTING THESE DATASETS

The datasets employed in our study represent the most prominent and extensively utilized benchmarks in this research area. Notably, MD17 and Motion Capture datasets serve as core evaluation benchmarks across multiple seminal works, including (Xu et al., 2024; Wu et al., 2024; Xu et al., 2023). Furthermore, the Adk equilibrium trajectory dataset has been established as a fundamental benchmark in key studies such as (Han et al., 2022b; Wu et al., 2024; Xu et al., 2024). This comprehensive dataset selection ensures that our experimental framework aligns with the current research domain and encompasses the most representative datasets in the field.

### C.2 IMPLEMENTATION DETAILS ON MD17 DATASET

The first column of Table 10 presents a unified set of hyper-parameters employed consistently across all experimental evaluations on the MD17 datasets. These parameters are uniformly applied to both our proposed EST model and all baseline methods. Both our EST and all other baselines are trained and tested on a single NVIDIA A100-80G GPU. The number of training, validation and testing sets are 500, 2000 and 2000, respectively.

In the MD17 dataset, each temporal graph contains up to 13 nodes. As for graph construction, We compute pairwise distances between all atoms, designating those within a distance threshold $\lambda$ as first-order neighbors. Each atom forms edges with both its first- and second-order neighbors to facilitate subsequent message passing.

### C.3 IMPLEMENTATION DETAILS ON MOTION CAPTURE

The third column of Table 10 presents a unified set of hyper-parameters employed consistently across all experimental evaluations on the motion capture datasets. These parameters are uniformly applied to both our proposed EST model and all baseline methods. Both our EST and all other baselines are

Table 10: Hyper-parameters of EST and other baseline methods. The temporal length $T$ denotes the length of initial timestamps, the time lag $\Delta t$ denotes the interval between two timestamps, the hidden size denotes the size of hidden states, and the layer denotes the number of layers.

| Hyper-parameter | MD17 | Protein | Motion Capture |
|---|---|---|---|
| Learning Rate | 5e-3 | 5e-5 | 5e-3 |
| Epochs | 500 | 500 | 500 |
| Temporal Length $T$ | 10 | 10 | 10 |
| Time Lag $\Delta t$ | 10 | 5 | 1 |
| Hidden Size | 16 | 16 | 16 |
| Layer | 2 | 2 | 2 |

trained and tested on a single NVIDIA A100-80G GPU. We adopt the setups and data splits from the official code of (Wu et al., 2024). The subject #35 (`Walk`) contains 1100/600/600 trajectories for training/validation/testing, while the subject #102 (`Basketball`) contains 600/300/300 trajectories for training/validation/testing.

In the Motion Capture dataset, each temporal graph contains up to 31 nodes. As for graph construction, directly connected joint nodes are classified as first-order neighbors. Each joint node establishes connections with both its first- and second-order neighbors to facilitate subsequent message passing. The invariant input features $h$ of all the joints are all 1s. In this experimental setup, we set $\Delta t = 1$, considering the substantial positional variations in motion data compared to MD17 and protein datasets. This choice mitigates the exponential growth of cumulative errors in autoregressive predictions. A larger $\Delta t$ would lead to unacceptably high error magnitudes across all methods as rollout steps increase, rendering the comparisons across different methods infeasible and compromising the validity of our comparative analysis.

### C.4 Implementation Details on Protein Dataset

The second column of Table 10 presents a unified set of hyper-parameters employed consistently across all experimental evaluations on the protein datasets. These parameters are uniformly applied to both our proposed EST model and all baseline methods. Both our EST and all other baselines are trained and tested on a single NVIDIA A100-80G GPU. The dataset is partitioned into training, validation, and test sets with a ratio of 6:2:2, respectively. This partition resulted in 2,482 samples for training, 827 for validation, and 827 for testing.

In the Protein dataset, each temporal graph contains 213 nodes. We construct a 4-channel equivariant 3D coordinate for the four backbone atoms (N, $C_\alpha$, C, O) , combining it with the corresponding atomic numbers as invariant input features $h$ to represent a single node in the graph. This node representation methodology is consistently applied across all baseline methods for comparative integrity. As for graph construction, we compute pairwise distances between all atoms, designating those within a distance threshold $\lambda$ as first-order neighbors. Each atom forms edges with its first-order neighbor to facilitate subsequent message passing.

## D  In-depth Analysis of the Dataset Results

### D.1 Predicted MSE Results Per Step

We conduct a reanalysis of Table 2 and calculate the per-frame prediction MSE difference between our approach and the previous state-of-the-art method ESTAG, as presented in Table 18. The findings reveal that the performance gap between our method and ESTAG widens progressively with increasing rollout steps, highlighting the enhanced capability of our model in longer-term prediction scenarios.

### D.2 Performance Improvement on Three Dataset

Our comprehensive empirical analysis substantiates the generalizability and robustness of our proposed EST across diverse experimental settings. The comparative evaluation, as delineated in Table 12

Table 11: Average per-frame prediction MSE difference between our method and the previous SOTA method.

| | Walk | | | Basketball | | |
|---|---|---|---|---|---|---|
| | R=10 | R=15 | R=20 | R=10 | R=15 | R=20 |
| ESTAG | 1.418 | 5.907 | 17.431 | 10.209 | 54.513 | 175.950 |
| EST-V | 1.095 | 4.084 | 12.708 | 9.658 | 49.374 | 148.988 |
| Reduction/step | -0.032 | -0.121 | -0.236 | -0.055 | -0.342 | -1.348 |

Table 12: Performance improvement compared to ESTAG on MD17 and Protein Dynamics

| | MD17 | | | Protein Dynamics | | |
|---|---|---|---|---|---|---|
| | R=10 | R=15 | R=20 | R=10 | R=15 | R=20 |
| MSE Reduction | -26.77% | -24.52% | -19.71% | -4.45% | -2.62% | -4.50% |

Table 13: Performance improvement compared to ESTAG on Motion Capture

| | Walk | | | Basketball | | |
|---|---|---|---|---|---|---|
| | R=10 | R=15 | R=20 | R=10 | R=15 | R=20 |
| MSE Reduction | -22.77% | -30.86% | -27.09% | -5.39% | -9.42% | -15.32% |

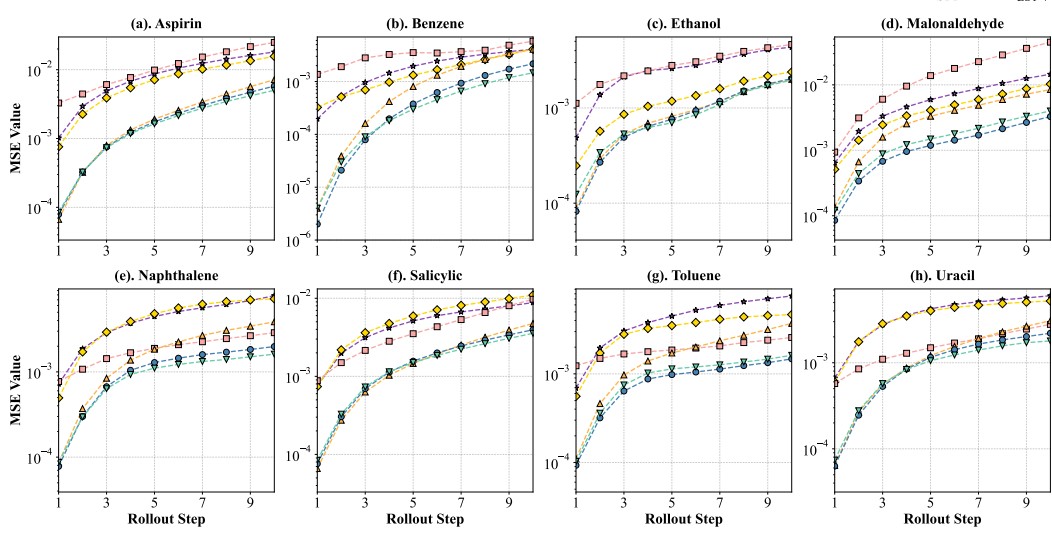

Figure 4: MSEs on MD17 up to 10 rollout steps (better zoom in).

and Table 13, encompasses multiple benchmarks with varying temporal horizons, where MD17 evaluations incorporate ensemble averaging across its 8 molecular constituents. The quantitative results consistently demonstrate significant performance enhancements relative to existing SOTA methods, regardless of the specific characteristics of the datasets or variations in rollout lengths.

Table 14: Prediction error ($\times 10^{-2}$) on MD17 dataset. Results averaged across 3 runs.

| | Aspirin | Benzene | Ethanol | Malonaldehyde | Naphthalene | Salicylic | Toluene | Uracil |
|---|---|---|---|---|---|---|---|---|
| ST_GNN-F | $9.403_{\pm 0.150}$ | $1.942_{\pm 0.086}$ | $2.650_{\pm 0.001}$ | $7.203_{\pm 0.102}$ | $4.311_{\pm 0.172}$ | $5.565_{\pm 0.251}$ | $4.530_{\pm 0.061}$ | $4.028_{\pm 0.374}$ |
| ST_GNN-V | $10.879_{\pm 0.001}$ | $2.174_{\pm 0.001}$ | $2.935_{\pm 0.001}$ | $9.134_{\pm 0.126}$ | $5.554_{\pm 0.020}$ | $7.035_{\pm 0.117}$ | $4.520_{\pm 0.140}$ | $4.626_{\pm 0.020}$ |
| ST_TFN-F | $7.974_{\pm 0.025}$ | $2.084_{\pm 0.001}$ | $2.441_{\pm 0.001}$ | $6.228_{\pm 0.066}$ | $4.768_{\pm 0.078}$ | $6.737_{\pm 0.024}$ | $4.041_{\pm 0.198}$ | $5.672_{\pm 0.098}$ |
| ST_TFN-V | $8.614_{\pm 0.027}$ | $2.270_{\pm 0.001}$ | $2.426_{\pm 0.001}$ | $7.247_{\pm 0.351}$ | $4.860_{\pm 0.057}$ | $6.870_{\pm 0.066}$ | $4.032_{\pm 0.143}$ | $5.623_{\pm 0.043}$ |
| STGCN-F | $8.079_{\pm 0.001}$ | $1.993_{\pm 0.004}$ | $2.786_{\pm 0.001}$ | $6.464_{\pm 0.001}$ | $5.829_{\pm 0.001}$ | $6.739_{\pm 0.001}$ | $4.724_{\pm 0.001}$ | $6.119_{\pm 0.001}$ |
| STGCN-V | $8.100_{\pm 0.001}$ | $2.240_{\pm 0.004}$ | $2.785_{\pm 0.001}$ | $6.467_{\pm 0.001}$ | $5.840_{\pm 0.001}$ | $6.976_{\pm 0.001}$ | $4.724_{\pm 0.001}$ | $6.175_{\pm 0.001}$ |
| ST_SE(3)-Tr.-F | $6.943_{\pm 0.082}$ | $2.085_{\pm 0.006}$ | $2.079_{\pm 0.001}$ | $5.775_{\pm 0.016}$ | $4.443_{\pm 0.046}$ | $5.577_{\pm 0.021}$ | $3.292_{\pm 0.004}$ | $4.914_{\pm 0.042}$ |
| ST_SE(3)-Tr.-V | $7.750_{\pm 0.062}$ | $2.232_{\pm 0.001}$ | $2.159_{\pm 0.001}$ | $6.810_{\pm 0.012}$ | $4.515_{\pm 0.019}$ | $6.063_{\pm 0.036}$ | $3.312_{\pm 0.012}$ | $5.185_{\pm 0.038}$ |
| ST_EGNN-F | $7.945_{\pm 0.040}$ | $3.764_{\pm 1.834}$ | $1.385_{\pm 0.001}$ | $4.661_{\pm 0.084}$ | $4.226_{\pm 0.752}$ | $6.214_{\pm 0.232}$ | $3.405_{\pm 0.178}$ | $3.303_{\pm 0.291}$ |
| ST_EGNN-V | $7.350_{\pm 0.589}$ | $1.922_{\pm 0.044}$ | $1.913_{\pm 0.001}$ | $5.183_{\pm 0.008}$ | $3.753_{\pm 0.145}$ | $5.536_{\pm 0.765}$ | $2.812_{\pm 0.109}$ | $3.845_{\pm 0.369}$ |
| ESTAG-F | $2.553_{\pm 0.414}$ | $1.524_{\pm 0.142}$ | $0.977_{\pm 0.001}$ | $2.758_{\pm 0.794}$ | $2.278_{\pm 0.211}$ | $2.239_{\pm 0.576}$ | $1.733_{\pm 0.591}$ | $1.600_{\pm 0.237}$ |
| ESTAG-V | $2.679_{\pm 0.193}$ | $1.699_{\pm 0.331}$ | $1.260_{\pm 0.001}$ | $3.098_{\pm 0.154}$ | $1.556_{\pm 0.126}$ | $2.406_{\pm 0.228}$ | $1.436_{\pm 0.223}$ | $1.805_{\pm 0.030}$ |
| EST-F | $\underline{2.345}_{\pm 0.077}$ | $\underline{0.873}_{\pm 0.098}$ | $\underline{0.968}_{\pm 0.001}$ | $\mathbf{1.442}_{\pm 0.025}$ | $\underline{1.297}_{\pm 0.185}$ | $\underline{1.895}_{\pm 0.034}$ | $\mathbf{0.957}_{\pm 0.117}$ | $\underline{1.470}_{\pm 0.234}$ |
| EST-V | $\mathbf{2.196}_{\pm 0.075}$ | $\mathbf{0.480}_{\pm 0.050}$ | $\mathbf{0.940}_{\pm 0.001}$ | $\underline{1.762}_{\pm 0.054}$ | $\mathbf{0.988}_{\pm 0.016}$ | $\mathbf{1.733}_{\pm 0.031}$ | $\underline{1.002}_{\pm 0.063}$ | $\mathbf{1.087}_{\pm 0.055}$ |

# E   MORE EXPERIMENT RESULTS ON MD17

## E.1   EACH STEP OF THE ROLLOUT PROCESS

Fig. 4 illustrates the MSE for various models over a 10-step rollout. It clearly demonstrates that while our model may not achieve optimal performance in the initial steps for some molecules, its advantages in mitigating cumulative errors become increasingly evident as the rollout progresses.

## E.2   VARIABLE-INPUT FOR OTHER BASELINES

In Table 1 of the main text, all baseline methods employ fixed-length historical frame inputs, potentially leading to unfair comparisons. To address this, we adapt most baseline methods to accept variable-length historical frame inputs, ensuring a more equitable evaluation. However, two methods were left unmodified: AGL-STAN, due to its predefined fixed-size convolutional layers, and Eqmotion, as modifying it would require substantial changes to its initial input, potentially significantly impacting model performance. Consequently, we revise all other baselines accordingly. Table 14 presents the detailed experimental results, where the suffix "F" denotes "fixed-length inputs" version, and "V" represents "variable-length inputs" version.

The experimental results reveal that when models are modified to accept variable-length inputs, ST_EGNN experiences performance degradation on some molecules, while the other four methods show varying degrees of performance decline across most molecules. This observation suggests that these methods struggle to efficiently process and utilize variable-length historical information, highlighting a limitation in their adaptability to more flexible input structures.

## E.3   EXPERIMENT ON MD17 WITH 15 AND 20 ROLLOUT STEPS

In addition to the results presented in the main text for the MD17 dataset with a rollout of 10 steps, we further investigate the model performance with extended rollout lengths of 15 and 20 steps. Tables 15 and 16 provide detailed results for these extended rollout scenarios. Analysis of these tables reveals that our proposed EST model maintains superior performance across the majority of molecules (8 / 8 for 15 rollout steps and 7 / 8 for 20 rollout steps) as the rollout length increases. Moreover, we observe that the gap in average prediction MSE between EST and other baseline models widens for some molecules as the rollout length increases. These findings provide compelling evidence of EST's excellent performance in longer-term physical dynamics simulation tasks, further validating its effectiveness and robustness in challenging predictive scenarios.

## E.4   EXPERIMENT ON MD17 WITH 100 ROLLOUT STEPS

Table 15: Predicted MSE ($\times 10^{-2}$) on MD17 dataset with 15 rollout steps. Results averaged across three trials.

| | Aspirin | Benzene | Ethanol | Malonaldehyde | Naphthalene | Salicylic | Toluene | Uracil |
|---|---|---|---|---|---|---|---|---|
| ST_GNN | $21.591_{\pm0.218}$ | $4.162_{\pm0.043}$ | $5.127_{\pm0.001}$ | $17.921_{\pm0.350}$ | $9.311_{\pm1.146}$ | $12.922_{\pm2.272}$ | $8.292_{\pm0.056}$ | $6.942_{\pm0.770}$ |
| ST_TFN | $18.264_{\pm0.050}$ | $4.729_{\pm0.001}$ | $4.436_{\pm0.075}$ | $15.429_{\pm0.301}$ | $9.301_{\pm0.138}$ | $14.429_{\pm0.205}$ | $6.528_{\pm0.112}$ | $11.733_{\pm0.148}$ |
| STGCN | $18.477_{\pm0.001}$ | $4.288_{\pm0.001}$ | $4.785_{\pm0.001}$ | $15.572_{\pm0.001}$ | $10.821_{\pm0.001}$ | $14.065_{\pm0.001}$ | $8.144_{\pm0.001}$ | $12.349_{\pm0.001}$ |
| ST_SE(3)-Tr. | $16.518_{\pm0.095}$ | $4.726_{\pm0.020}$ | $3.796_{\pm0.001}$ | $14.615_{\pm0.025}$ | $8.321_{\pm0.041}$ | $12.071_{\pm0.098}$ | $5.796_{\pm0.123}$ | $10.286_{\pm0.086}$ |
| ST_EGNN | $19.298_{\pm0.551}$ | $85.91_{\pm60.391}$ | $3.455_{\pm0.001}$ | $11.984_{\pm0.737}$ | $10.95_{\pm1.698}$ | $12.898_{\pm1.196}$ | $6.061_{\pm0.442}$ | $6.575_{\pm0.769}$ |
| AGL-STAN | $31.171_{\pm1.348}$ | $7.054_{\pm0.196}$ | $5.834_{\pm0.194}$ | $61.618_{\pm3.265}$ | $3.493_{\pm0.231}$ | $18.208_{\pm1.420}$ | $3.331_{\pm0.096}$ | $3.646_{\pm0.293}$ |
| Eqmotion | $28.426_{\pm0.001}$ | $41.291_{\pm0.001}$ | $32.310_{\pm0.001}$ | $24.564_{\pm0.001}$ | $18.694_{\pm0.001}$ | $23.711_{\pm0.001}$ | $12.649_{\pm0.001}$ | $17.329_{\pm0.001}$ |
| ESTAG | $8.801_{\pm1.030}$ | $3.001_{\pm0.515}$ | $2.584_{\pm0.001}$ | $5.648_{\pm0.771}$ | $2.968_{\pm1.007}$ | $7.109_{\pm0.724}$ | $3.473_{\pm1.153}$ | $4.713_{\pm0.475}$ |
| EST-F | $\underline{7.071}_{\pm0.617}$ | $\underline{2.672}_{\pm0.292}$ | $\mathbf{2.485}_{\pm0.001}$ | $\underline{5.549}_{\pm0.126}$ | $\mathbf{2.645}_{\pm0.204}$ | $\underline{5.468}_{\pm0.465}$ | $\underline{2.495}_{\pm0.410}$ | $\underline{3.056}_{\pm0.839}$ |
| EST-V | $\mathbf{6.881}_{\pm0.755}$ | $\mathbf{1.905}_{\pm0.009}$ | $\underline{2.551}_{\pm0.001}$ | $\mathbf{4.705}_{\pm0.107}$ | $\underline{2.784}_{\pm0.432}$ | $\mathbf{5.162}_{\pm0.220}$ | $\mathbf{2.166}_{\pm0.251}$ | $\mathbf{2.419}_{\pm0.208}$ |

Table 16: Predicted MSE ($\times 10^{-2}$) on MD17 dataset with 20 rollout steps. Results averaged across three trials.

| | Aspirin | Benzene | Ethanol | Malonaldehyde | Naphthalene | Salicylic | Toluene | Uracil |
|---|---|---|---|---|---|---|---|---|
| ST_GNN | $39.298_{\pm1.003}$ | $\underline{6.909}_{\pm0.310}$ | $8.101_{\pm0.001}$ | $35.272_{\pm0.565}$ | $24.729_{\pm4.049}$ | $18.066_{\pm1.511}$ | $13.540_{\pm0.708}$ | $10.506_{\pm0.976}$ |
| ST_TFN | $33.855_{\pm0.097}$ | $8.378_{\pm0.021}$ | $6.561_{\pm0.023}$ | $30.603_{\pm0.098}$ | $13.525_{\pm0.285}$ | $24.963_{\pm0.051}$ | $10.024_{\pm1.426}$ | $20.743_{\pm0.186}$ |
| STGCN | $33.887_{\pm0.001}$ | $6.391_{\pm0.001}$ | $6.993_{\pm0.001}$ | $30.293_{\pm0.001}$ | $17.026_{\pm0.001}$ | $23.247_{\pm0.001}$ | $11.709_{\pm0.001}$ | $20.991_{\pm0.001}$ |
| ST_SE(3)-Tr. | $30.851_{\pm0.093}$ | $8.313_{\pm0.011}$ | $5.862_{\pm0.001}$ | $28.374_{\pm0.047}$ | $13.070_{\pm0.145}$ | $20.501_{\pm0.327}$ | $13.673_{\pm7.056}$ | $17.507_{\pm0.058}$ |
| ST_EGNN | $46.452_{\pm5.563}$ | $39.574_{\pm7.444}$ | $5.225_{\pm0.001}$ | $23.438_{\pm0.590}$ | $14.064_{\pm0.240}$ | $23.724_{\pm1.444}$ | $9.006_{\pm0.962}$ | $10.560_{\pm1.665}$ |
| AGL-STAN | $61.723_{\pm9.104}$ | $22.043_{\pm3.660}$ | $10.389_{\pm0.686}$ | $158.11_{\pm1.651}$ | $8.398_{\pm1.199}$ | $23.458_{\pm3.255}$ | $4.947_{\pm0.289}$ | $8.064_{\pm2.291}$ |
| Eqmotion | $191.846_{\pm0.001}$ | $801.551_{\pm0.001}$ | $836.737_{\pm0.001}$ | $103.339_{\pm0.001}$ | $87.615_{\pm0.001}$ | $113.888_{\pm0.001}$ | $135.910_{\pm0.001}$ | $144.932_{\pm0.001}$ |
| ESTAG | $21.068_{\pm1.472}$ | $7.377_{\pm1.410}$ | $\mathbf{4.317}_{\pm0.001}$ | $13.538_{\pm2.102}$ | $7.124_{\pm1.962}$ | $12.668_{\pm2.574}$ | $5.270_{\pm0.614}$ | $7.323_{\pm0.635}$ |
| EST-F | $\mathbf{16.527}_{\pm0.101}$ | $7.287_{\pm0.456}$ | $\underline{4.582}_{\pm0.001}$ | $\underline{11.631}_{\pm1.080}$ | $\underline{6.823}_{\pm2.132}$ | $\underline{11.867}_{\pm3.392}$ | $\mathbf{3.050}_{\pm0.428}$ | $\underline{4.963}_{\pm0.272}$ |
| EST-V | $\underline{17.405}_{\pm1.891}$ | $\mathbf{5.376}_{\pm0.501}$ | $5.028_{\pm0.001}$ | $\mathbf{11.448}_{\pm1.120}$ | $\mathbf{5.617}_{\pm2.350}$ | $10.372_{\pm0.511}$ | $\underline{3.288}_{\pm0.130}$ | $\mathbf{4.588}_{\pm0.794}$ |

Conducting long-term rollouts for physical dynamics simulations is inherently a challenging task. Considering the characteristics of the datasets, such as the highly oscillatory dynamics in MD17, a rollout length of 20 steps is already an effective and reasonable choice for evaluating model performance. For reference, previous methods such as ESTAG use only 10-step rollouts. To further extend our analysis, we explored the model predictions using 100 rollout steps. The results are presented in Table 17. The results show that our method still consistently outperforms ESTAG. However, after 100 rollout steps, the model's MSE deteriorates from 1e-5 to 1e-3 in magnitude. This highlights that, due to the dramatic dynamics of MD17, achieving stable rollouts over 100 steps remains extremely challenging.

Table 17: Results for 100-step rollout results on MD17.

| Method | R=1 | R=25 | R=50 | R=75 | R=100 |
|---|---|---|---|---|---|
| ESTAG | 0.090 | 5.969 | 6.637 | 6.378 | 6.406 |
| EST | **0.078** | **4.181** | **5.343** | **6.163** | **5.993** |

### E.5 GENERALIZATION FOR CONTINUOUS-TIME PREDICTION

To verify whether our model generalizes to continuous-time prediction (e.g., irregular timesteps), we conduct an experiment where models are trained on MD17 with a fixed sampling interval ($\Delta t = 10$) but tested on random intervals sampled from the range $[1, 30]$ (e.g., intervals of 15, 5, or 20 frames). We compare EST with the previous SOTA model, ESTAG, and the results are presented in Table 19 (suffixed with -irre). The data reveals that ESTAG suffers severe performance degradation under irregular timesteps; for instance, the error on Benzene spikes from 1.524 to 14.806 (a nearly 9-fold increase), indicating that traditional methods rely heavily on a fixed $\Delta t$ and fail to learn the continuous dynamics equation. In contrast, EST demonstrates remarkable stability. Despite the

Table 18: Average per-frame prediction MSE difference between our method and the previous SOTA method.

| | Walk | | | Basketball | | |
|---|---|---|---|---|---|---|
| | R=10 | R=15 | R=20 | R=10 | R=15 | R=20 |
| ESTAG | 1.418 | 5.907 | 17.431 | 10.209 | 54.513 | 175.950 |
| EST-V | 1.095 | 4.084 | 12.708 | 9.658 | 49.374 | 148.988 |
| Reduction/step | -0.032 | -0.121 | -0.236 | -0.055 | -0.342 | -1.348 |

significant increase in task difficulty, EST maintains low error levels under irregular sampling, outperforming ESTAG across all molecules. This indicates that EST, through the TDG module's differential modeling and the Transformer's attention mechanism, adaptively adjusts prediction magnitudes based on implicit time spans, thereby naturally generalizing to continuous or irregular prediction tasks.

Table 19: Performance under irregular timesteps on MD17.

| Method | Aspirin | Benzene | Ethanol | Malonaldehyde | Naphthalene | Salicylic | Toluene | Uracil |
|---|---|---|---|---|---|---|---|---|
| ESTAG | 2.553±0.414 | 1.524±0.142 | 0.977±0.001 | 2.758±0.794 | 2.278±0.211 | 2.239±0.576 | 1.733±0.591 | 1.600±0.237 |
| ESTAG-irre | 10.861±1.175 | 14.806±0.282 | 1.650±0.001 | 4.196±0.230 | 5.509±0.339 | 5.060±0.246 | 5.704±2.933 | 4.468±0.533 |
| EST | 2.196±0.075 | 0.480±0.050 | 0.940±0.001 | 1.762±0.054 | 0.988±0.016 | 1.733±0.031 | 1.002±0.063 | 1.087±0.055 |
| EST-irre | 3.679±0.128 | 0.875±0.009 | 1.488±0.001 | 2.677±0.046 | 1.874±0.031 | 3.434±0.084 | 2.041±0.103 | 2.205±0.024 |

### E.6 GENERALIZATION UNDER RAPIDLY CHANGING SCENARIOS

To validate the model's ability to predict unseen rapid changes, we design an experiment where models are trained on data with a fixed interval of $\Delta t = 10$ but tested on a larger interval of $\Delta t = 15$. This represents a 50% increase in particle displacement within the same physical time, constituting a "rapid change" unseen during training. We compare EST against ESTAG, with results shown in Table 20 (suffixed with rapid-change). As indicated in the table, ESTAG exhibits catastrophic failure when facing these unseen rapid changes; for example, the error on Benzene increases nearly 15-fold (from 1.524 to 23.808), and the error on Naphthalene increases nearly 7-fold. This suggests that traditional models severely overfit the motion magnitude observed during training. Conversely, EST demonstrates exceptional robustness. On Benzene, despite the 50% increase in magnitude, the error only rises to 1.510, which outperforms ESTAG's performance on the standard test set. On complex molecules like Naphthalene and Salicylic, EST's error is only 1/5 to 1/3 of ESTAG's. This capability stems from EST's modeling of differences and the long-range attention mechanism; the model learns trends of change rather than memorizing positions, enabling effective extrapolation when the magnitude ($\Delta t$) changes.

Table 20: Performance under rapid change scenarios on MD17.

| Method | Aspirin | Benzene | Ethanol | Malonaldehyde | Naphthalene | Salicylic | Toluene | Uracil |
|---|---|---|---|---|---|---|---|---|
| ESTAG | 2.553±0.414 | 1.524±0.142 | 0.977±0.001 | 2.758±0.794 | 2.278±0.211 | 2.239±0.576 | 1.733±0.591 | 1.600±0.237 |
| ESTAG-rapid-change | 13.887±1.615 | 23.808±1.688 | 3.947±0.001 | 7.626±0.279 | 17.884±0.889 | 15.042±0.746 | 6.932±0.061 | 7.690±1.092 |
| EST | 2.196±0.075 | 0.480±0.050 | 0.940±0.001 | 1.762±0.054 | 0.988±0.016 | 1.733±0.031 | 1.002±0.063 | 1.087±0.055 |
| EST-rapid-change | 6.547±0.104 | 1.510±0.168 | 4.007±0.001 | 5.342±0.083 | 3.727±0.101 | 6.161±0.490 | 4.272±0.363 | 4.271±0.141 |

### E.7 GENERALIZING TO SYSTEMS WITH VARYING COMPONENTS AND INTERACTIONS

To further validate our model's ability to generalize to systems with different numbers of components or interaction types, we conduct the following experiment. We select one molecule as the unseen test target at a time and combine the datasets of the remaining 7 molecules as the training set. Since the molecules in MD17 exhibit significant differences in atomic counts and chemical structures (interaction types), this setup rigorously tests the model's generalization capability across systems

with varying component numbers and interaction types. We compare our EST model with the previous SOTA model, ESTAG, under this generalization setting (labeled as -generalize). The results are presented in Table 21. Our EST model outperforms ESTAG on all molecules in this generalization setting. On average, the error of EST-generalize (2.760) is significantly lower than that of ESTAG-generalize (3.527). This demonstrates that our architecture captures universal physical dynamic laws more effectively than previous methods and successfully generalizes to systems with different numbers of components.

Table 21: Performance generalization capabilities on MD17.

| Method | Aspirin | Benzene | Ethanol | Malonaldehyde | Naphthalene | Salicylic | Toluene | Uracil | Average |
|---|---|---|---|---|---|---|---|---|---|
| ESTAG | 2.553±0.414 | 1.524±0.142 | 0.977±0.001 | 2.758±0.794 | 2.278±0.211 | 2.239±0.576 | 1.733±0.591 | 1.600±0.237 | 1.958 |
| ESTAG-generalize | 5.037±0.838 | 2.080±0.177 | 2.763±0.085 | 5.823±0.195 | 3.099±0.390 | 2.791±0.094 | 3.515±0.370 | 3.114±0.179 | 3.527 |
| EST | 2.196±0.075 | 0.480±0.050 | 0.940±0.001 | 1.762±0.054 | 0.988±0.016 | 1.733±0.031 | 1.002±0.063 | 1.087±0.055 | 1.273 |
| EST-generalize | 3.747±0.383 | 1.685±0.188 | 2.613±0.181 | 5.756±0.262 | 1.578±0.021 | 1.952±0.124 | 2.607±0.059 | 2.148±0.154 | 2.760 |

## E.8 COMPLEXITY ANALYSIS

The inference time of different methods are comprehensively analyzed in Table 23. The computational complexity inherent in Transformer-based architectures exhibits quadratic growth with respect to temporal sequence length $O(T^2)$. However, in our experiments, where temporal sequences are limited to 10-20 steps, the additional computational cost remains negligible. It is crucial to clarify that, within the current experimental settings (short sequences of T=10 to 20), our claim is solely that the additional computational cost from this quadratic growth is "negligible". This does not suggest that EST's total runtime is insignificant compared to non-Transformer baselines. Due to the introduction of Transformer architecture, EST is indeed slower than non-Transformer architectures. Although EST-V (0.136s) is approximately 4 times slower than ESTAG (0.031s), we achieve a qualitative breakthrough in prediction accuracy. For instance, on Benzene and Naphthalene, the errors drop from ESTAG's 1.524 and 2.278 to 0.480 and 0.988, respectively (reductions of approximately 68% and 56%). On more complex systems like Protein and Motion Capture, our long-range rollout errors ($R = 20$) are also significantly lower than ESTAG's. This substantial reduction in error is crucial for high-fidelity physical simulation. Furthermore, when compared to Eqmotion, EST-V (0.136s) is in the same order of magnitude as EqMotion (0.173s) but far outperforms it. Therefore, we believe the enhanced model efficacy outweighs the computational overhead.

For larger systems, our current model indeed faces computational scalability challenges as graph size and sequence length increase. To address this, we plan to explore acceleration techniques such as the virtual node paradigm introduced in FastEGNN (Zhang et al., 2024). Potential implementations could involve introducing virtual nodes along both temporal and spatial dimensions. A key point would be determining the optimal configuration—whether to employ per-timestep virtual nodes or a globally shared virtual node across all timesteps. This direction appears particularly promising for balancing efficiency and performance. Notably, our model can further benefit from efficient techniques like flash attention (Dao et al., 2022), widely adopted in NLP for reducing computational complexity. It can be seamlessly integrated for acceleration.

## E.9 ALL ATOM PROTEIN DYNAMICS

We have conducted all-atom experiments in protein dynamics simulation, the results are shown in Table 22. Notably, our EST method still demonstrates significant performance advantages over competing approaches, indicating its robust capability in handling macromolecular.

Table 22: Results on all-atom protein dynamics.

| | EGNN | ESTAG | EST |
|---|---|---|---|
| R=10 | 2.531 | 2.533 | **2.430** |

## F VISUALIZATION OF THE ATTENTION MAP

In Fig. 5, we present the visualization of the attention map for EST-V. The visualization shows that most timesteps attend to a significant portion of the historical frames. This further validates that EST-V effectively captures and integrates historical input information to enhance prediction performance.

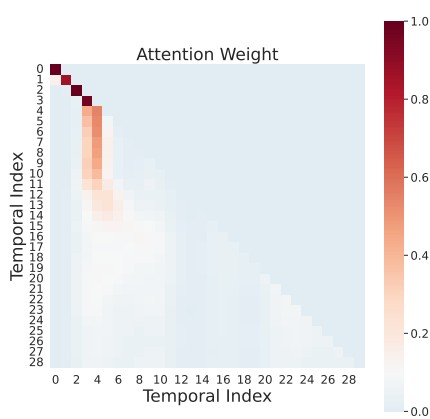

Figure 5: Visualization of the attention map.

Table 23: Computational efficiency across different methods.

|  | runtimes(s) |
| --- | --- |
| ST_GNN | 0.048 |
| ST_TFN | 21.101 |
| ST_GCN | 0.014 |
| ST_SE(3)_Tr. | 46.590 |
| ST_EGNN | 0.013 |
| AGL-STAN | 0.075 |
| Eqmotion | 0.173 |
| ESTAG | 0.031 |
| EST-shared para | 0.106 |
| EST-V | 0.136 |

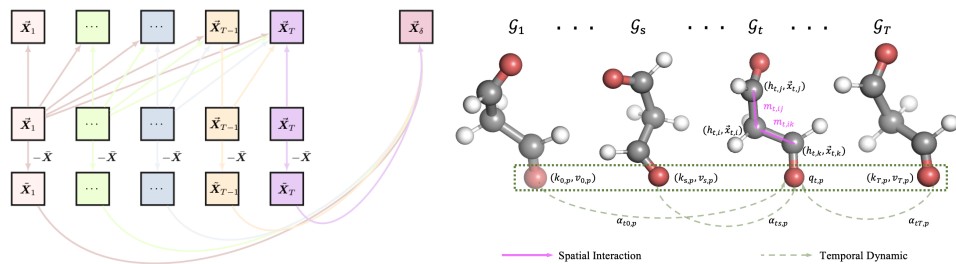

(a) How TDG works within the proposed model.

(b) Diagrams for Equivariant Spatio-Temporal Block.

Figure 6: (a) TDG within the proposed model; (b) Equivariant Spatio-Temporal Block.

# G  LLM USAGE

Large language models (LLMs) were used to assist in the writing and polishing of this manuscript. Specifically, an LLM was employed to help refine the language, improve readability, and improve clarity in various sections of the document. It supported tasks such as rephrasing sentences, checking grammar, and improving the overall flow of the text.

It is important to note that the LLM was not involved in the generation of ideas, research methodology, or experimental design. All research concepts, analyses, and conclusions were developed and conducted by the authors. The contributions of the LLM were strictly limited to improving the linguistic quality of the manuscript and did not extend to the scientific content or data analysis.

The authors assume full responsibility for the entire content of the manuscript, including any text generated or refined with LLM assistance. We have verified that all LLM-assisted writing adheres to ethical standards and does not constitute plagiarism or scientific misconduct.

