# OpenReview forum: "Physical Dynamics as Next Geometric Graph Prediction"
_ICLR.cc/2026/Conference — Submitted to ICLR 2026_

### Official Review · Reviewer_HooZ · 2025-10-17

**Soundness:** 3
**Presentation:** 3
**Contribution:** 3
**Rating:** 6
**Confidence:** 2

**Summary:**

This paper addresses the problem of simulating physical dynamics by framing it as an autoregressive prediction of spatiotemporal graph sequences. The authors propose the Equivariant Spatiotemporal Transformer (EST), a novel encoder-decoder architecture that maintains E(3) symmetries. The model incorporates specialized spatiotemporal blocks that alternate between spatial message passing and temporal self-attention. A key contribution is the Temporal Difference Graph (TDG), a module initialized from frame-wise differences that interacts with the historical trajectory to model global dynamics and mitigate cumulative error, a persistent challenge in autoregressive simulation. The authors evaluate EST on molecular, protein, and human motion dynamics datasets, demonstrating state-of-the-art performance, particularly in long-horizon predictions.

**Strengths:**

1. The introduction of the Temporal Difference Graph (TDG) is a creative and promising approach to the critical problem of error accumulation in autoregressive models (see Sec. 1, Sec. 3.2). This moves beyond simple frame-to-frame prediction and introduces a mechanism to explicitly model system dynamics over time.
2. The experiments are extensive, spanning three distinct and relevant physical scales—molecular (MD17), human motion (Motion Capture), and protein dynamics (Adk trajectory)—which convincingly demonstrates the model's versatility and generalizability (see Sec. 4).
3. The ablation study presented in Table 4 systematically deconstructs the model to validate the contribution of its key components. These experiments confirm the importance of maintaining equivariance, the efficacy of the TDG, the benefit of the encoder-decoder structure, and the rationale for using causal attention in the encoder (see Sec. 4.4).

**Weaknesses:**

1. The paper claims the TDG helps circumvent cumulative errors, but the theoretical justification provided (Taylor expansion in Sec. B.6.1) is a general argument for predicting velocity rather than position. It does not fully explain why the specific implementation of the TDG—as an independent graph token that interacts with all historical frames via attention—is superior to simpler or more direct velocity prediction schemes used in prior work.
2. The paper downplays the computational cost, stating that for short sequences the overhead is "negligible" (see Sec. E.4 ). However, the runtime analysis in Table 17 shows that EST-V is approximately 4 times slower than the previous SOTA (ESTAG) and 10 times slower than the fastest baseline (ST_EGNN). This is a significant trade-off that deserves a more prominent and nuanced discussion in the main text.
3. Why is predicting the difference from the previous frame simpler than predicting the next frame directly? How can this be demonstrated?
4. In the description of Eq. (4), please clarify whether the mean reduction term $\overline{x}$ is computed only once based on the initial input trajectory or is dynamically recomputed at each step of the autoregressive rollout process.

**Questions:**

Reference weaknesses.

---

> ### Author Response · Authors · 2025-11-24
> **Responses to Reviewer HooZ (Part 1/2)**
>
> We sincerely thank you for the time and careful consideration you have given to providing detailed and constructive feedback. Your valuable insights have greatly improved both the technical accuracy and clarity of our manuscript. We have meticulously revised the paper to incorporate your suggestions. Below, we respond to each of your comments point by point.
>
> > **Weakness1: It does not fully explain why the specific implementation of the TDG—as an independent graph token that interacts with all historical frames via attention—is superior to simpler or more direct velocity prediction schemes used in prior work.**
>
> Thank you for your detailed observation. The core reason for designing the TDG as an independent interaction graph token is the need to capture "global dynamics evolution trends" rather than merely calculating "local instantaneous velocity". Traditional simple velocity prediction schemes (such as GNS) typically rely on only the most recent frames to calculate current acceleration or velocity, which is a local operation. In contrast, although the TDG is initialized as the difference between the last two frames, upon entering the decoder, it interacts with all historical frames via the attention mechanism. This implies that TDG serves not merely as a predictive target, but also as a dynamic learner. It aggregates long-range dependency information from the entire historical trajectory (e.g., the oscillation cycle of a molecule or the long-term momentum trend of a macroscopic object) to dynamically refine the current update.
>
> To demonstrate that this interactive TDG is superior to simple velocity prediction, we directly compared it with the GNS model, which employs simple velocity/acceleration prediction strategies, in Appendix B.9. The experimental results, shown in Table S9, indicate that the error of EST on the MD17 dataset is an order of magnitude lower than that of GNS (e.g., Aspirin: 2.196 vs 23.545). This substantial performance gap proves that merely predicting velocity is insufficient; refining this difference prediction through TDG's global interaction mechanism is the key factor.
>
> **Table S9: Performance between EST and GNS on MD17.**
> | Method | Aspirin      | Benzene     | Ethanol      | Malonaldehyde | Naphthalene  | Salicylic    | Toluene      | Uracil       |
> | :----- | :----------- | :---------- | :----------- | :------------ | :----------- | :----------- | :----------- | :----------- |
> | GNS    | 23.545±0.001 | 2.53±0.002  | 15.793±0.001 | 17.878±0.001  | 12.768±0.006 | 22.498±0.001 | 12.109±0.001 | 21.299±0.001 |
> | EST    | 2.196±0.075  | 0.480±0.050 | 0.940±0.001  | 1.762±0.054   | 0.988±0.016  | 1.733±0.031  | 1.002±0.063  | 1.087±0.055  |
>
> > **Weakness2: The paper downplays the computational cost, stating that for short sequences the overhead is "negligible" (see Sec. E.4 ). However, the runtime analysis in Table 17 shows that EST-V is approximately 4 times slower than the previous SOTA (ESTAG) and 10 times slower than the fastest baseline (ST_EGNN). This is a significant trade-off that deserves a more prominent and nuanced discussion in the main text.**
>
> Thank you for your suggestion. The context for the term "negligible" in Section E.4 is that the computational complexity of the Transformer architecture grows quadratically with the time series length ($O(T^2)$). Our point is that within the current experimental settings (short sequences where $T=10$ to $20$), the additional computational cost incurred by this quadratic growth is "negligible", rather than implying that EST's total runtime is insignificant compared to non-Transformer baselines. Due to the introduction of Transformer architecture, EST is indeed slower than non-Transformer architectures.
>
> Although EST-V (0.136s) is approximately 4 times slower than ESTAG (0.031s), we achieve a qualitative breakthrough in prediction accuracy. For instance, on Benzene and Naphthalene, the errors drop from ESTAG's 1.524 and 2.278 to 0.480 and 0.988, respectively (reductions of approximately 68% and 56%). On more complex systems like Protein and Motion Capture, our long-range rollout errors ($R=20$) are also significantly lower than ESTAG's. This substantial reduction in error is crucial for high-fidelity physical simulation. Furthermore, when compared to Eqmotion, EST-V (0.136s) is in the same order of magnitude as EqMotion (0.173s) but far outperforms it. Therefore, we believe the enhanced model efficacy outweighs the computational overhead.
>
> We also explicitly discuss acceleration schemes for larger-scale systems in the main text. For scenarios where speed is critical, we propose future directions such as introducing flash attention [A] or virtual nodes [B] technologies, which can significantly reduce computational complexity while maintaining the Transformer's expressive power. We have discussed this trade-off more prominently in the revised version.

---

> ### Author Response · Authors · 2025-11-24
> **Responses to Reviewer HooZ (Part 2/2)**
>
> > **Weakness3: Why is predicting the difference from the previous frame simpler than predicting the next frame directly? How can this be demonstrated?**
>
> Thank you for your valuable question. We explain this from three aspects. First, in physical dynamics, time steps are typically short, leading to strong local correlations between adjacent frames. This means $\vec{X}\_{T+1}$ is highly similar to $\vec{X}\_{T}$. If the model is tasked with directly predicting $\vec{X}\_{T+1}$, it must first learn to "copy" the majority of information from $\vec{X}\_{T}$ (i.e., learn an identity mapping) and then learn the minute change $\Delta \vec{X}$. Learning a perfect identity mapping is difficult and inefficient for deep networks. Conversely, if the model is tasked with predicting the difference  $\Delta \vec{X}$ (as done by the TDG in EST), the task becomes much simpler.  $\vec{X}\_{T}$ is preserved as a shortcut via the final addition $\vec{X}\_{T+1}=\vec{X}\_{T}+\Delta \vec{X}$. Second, the difference $\Delta \vec{X}$ is typically a sparser signal closer to zero compared to the absolute coordinates $\vec{X}\_{T+1}$. For neural networks, learning to predict small fluctuations near zero is generally easier than predicting large absolute values potentially far from the origin. Finally, this hypothesis is strongly supported by our ablation studies. In Table 4 of the original paper, we show the results for "w/o TDG". This variant leads to a significant performance drop. This demonstrates that weakening this explicit differential/dynamic modeling mechanism severely impairs the model's ability to capture physical trajectories.
>
> > **Weakness4: In the description of Eq. (4), please clarify whether the mean reduction term $\bar{x}$ is computed only once based on the initial input trajectory or is dynamically recomputed at each step of the autoregressive rollout process.**
>
> Thank you for your suggestion, and we apologize for the lack of clarity in our description. The mean reduction term $\bar{x}$ is computed only once based on the initial input trajectory and is not dynamically recomputed at each step of the autoregressive rollout process. We will clearly describe this point in the revised manuscript.
>
> Finally, we sincerely thank the reviewer for the opportunity to further clarify the motivation behind our TDG design. We are also grateful for the valuable suggestions to revise and supplement the complexity analysis and the formulas description. These revisions have significantly enhanced the readability of our paper and made the overall presentation more comprehensive.
>
> [A] Flashattention: Fast and memory-efficient exact attention with io-awareness. NIPS 2022.
>
> [B] Improving equivariant graph neural networks on large geometric graphs via virtual nodes learning. ICML 2024.

---

> ### Author Response · Authors · 2025-11-27
> **Looking forward to your responses**
>
> Dear Reviewer HooZ,
>
> We sincerely appreciate the time and effort you have dedicated to reviewing our paper. We have made every effort to address your questions and concerns in detail, and have updated the manuscript accordingly (with changes highlighted in red). If you find our responses satisfactory, we would be grateful for your reconsideration of the score!
>
> Best regards,
>
> The Authors

---

### Official Review · Reviewer_isjy · 2025-10-26

**Soundness:** 3
**Presentation:** 4
**Contribution:** 3
**Rating:** 4
**Confidence:** 2

**Summary:**

This paper proposes a novel approach to modeling physical dynamics by casting it as a next-step geometric graph prediction problem. Instead of representing physical systems as sequences of state vectors, the authors model each timestep as a graph and use geometric GNNs to predict the evolution of these graphs over time. The approach aims to integrate structural inductive biases with learned dynamics, leveraging the flexibility of GNNs in representing complex interactions and spatial relationships. The method is evaluated on several physical simulation tasks and is compared against existing neural and physics-based baselines.

**Strengths:**

1. The idea of formulating physical dynamics as a geometric graph prediction problem is conceptually interesting, and could potentially offer a unified framework for structured dynamical modeling.

2. The paper leverages geometric deep learning techniques, such as equivariant GNNs, which are well-suited to modeling the symmetries inherent in physical systems.

**Weaknesses:**

1. While the overall framework is promising, the technical details are somewhat underdeveloped. Key components (e.g., how graphs are constructed at each timestep, how node/edge features evolve) are described at a high level and lack sufficient mathematical clarity or justification.

2. The experimental results look weak. Only a small number of benchmarks are used, and comparisons with strong recent baselines (e.g., learned simulators like GNS or differentiable physics engines) are missing.

3. The model’s ability to generalize to systems with different numbers of components or interaction types is not convincingly demonstrated, which is a critical capability for physical simulation models.

**Questions:**

Pls refer to weaknesses

---

> ### Author Response · Authors · 2025-11-24
> **Responses to Reviewer isjy (Part 1/3)**
>
> We are deeply grateful for the time and thoughtful effort you have dedicated to offering comprehensive and valuable feedback. Your insightful suggestions have been instrumental in enhancing both the technical rigor and readability of our manuscript. We have thoroughly revised the paper to address all your comments. Below, we respond to each of your points in detail.
>
> > **Weakness1: While the overall framework is promising, the technical details are somewhat underdeveloped. Key components (e.g., how graphs are constructed at each timestep, how node/edge features evolve) are described at a high level and lack sufficient mathematical clarity or justification.**
>
> Thank you for your feedback. We provide a detailed clarification of the relevant technical details here.
>
> **Regarding "how graphs are constructed at each timestep":** For all timesteps except the Temporal Difference Graph ($\mathcal{G}\_\delta$), we treat the scene objects as graph structures and model them using EGNN. Depending on the task, we treat atoms, human skeleton joints, and residues as graph nodes for molecular dynamics, motion capture, and protein dynamics, respectively. Table S6 briefly illustrates the feature information on the graphs for each dataset. For more detailed graph construction, we provide comprehensive descriptions in Appendices C.2, C.3, and C.4. For the detailed construction process of the Temporal Difference Graph $\mathcal{G}_\delta$, specific descriptions are available in the "Temporal Difference Graph" paragraph within the "Equivariant Decoder" module in Section 3.2.
>
> **Table S6:  The feature information on the graphs for each dataset.**
> |                  | Node feature                          | Edge connection                   |
> | :--------------- | :------------------------------------ | :-------------------------------- |
> | MD17             | atomic numbers                        | first- and second-order neighbors |
> | Motion Capture   | joint nodes                           | first- and second-order neighbors |
> | Protein Dynamics | atomic numbers of four backbone atoms | first-order neighbor              |
>
> **Regarding "how node/edge features evolve":** We introduce this process in detail in Section 3.2, divided into the Equivariant Encoder and Equivariant Decoder modules. We now provide a brief overview of how node and edge features evolve. In the Equivariant Encoder, Equation 2 represents the update process of node/edge features on the graph corresponding to each timestep (temporal graph) using EGNN in the spatial dimension. Equation 3 represents the update of these features in the temporal dimension using the equivariant self-attention mechanism. In the Equivariant Decoder, we update the node/edge features of every temporal graph (excluding $\mathcal{G}_\delta$) in the input data using the same methods as Equation 2 and Equation 3, sequentially in spatial and temporal dimensions. We omitted the repetitive description of this step in the original paper for brevity. Furthermore, due to the specificity of the TDG, we describe the update of node/edge features on the TDG in the temporal dimension separately using Equation 4. Finally, Equation 5 represents the process where we generate Queries from all temporal graphs updated by the Equivariant Decoder, and Keys and Values from those updated by the Equivariant Encoder, applying the equivariant cross-attention mechanism to update all node/edge features in the Equivariant Decoder.
>
> Detailed descriptions and explanations for the above equations are available in Section 3.2. In addition to Figure 2, which illustrates the overall EST process, we provide further visualizations in Figures 6(a) and 6(b) in the Appendix to intuitively demonstrate how the TDG module interacts with other temporal graphs and how the model updates in spatial and temporal dimensions. Finally, the proof of equivariance for the overall EST framework is available in Appendix A.

---

> ### Author Response · Authors · 2025-11-24
> **Responses to Reviewer isjy (Part 2/3)**
>
> > **Weakness2: The experimental results look weak. Only a small number of benchmarks are used, and comparisons with strong recent baselines (e.g., learned simulators like GNS or differentiable physics engines) are missing.**
>
> Thank you for your suggestion. **Regarding "Only a small number of benchmarks are used":** The three datasets listed in our paper are widely recognized and popular in this field. Specifically, MD17 and Motion Capture are used as primary experimental datasets in [B, C, D], while the Adk equilibrium trajectory dataset is also used as a primary dataset in [A, B, D]. Therefore, our selection covers the essential and commonly used datasets in the current research domain as comprehensively as possible.
>
> **Regarding "comparisons with strong recent baselines are missing":** As you mentioned, GNS is a significant baseline in this field. We already included a detailed comparison with GNS in Appendix B.9 (Table 8), and the results are also shown in Table S7 below. The results indicate that our method significantly outperforms GNS, highlighting its superiority. We also reported the results of Equiformer [E], an advanced and widely recognized method in this field, in Table 6 of the Appendix, which are also displayed in Table S7. While Equiformer exhibits performance gains over GNS, its performance still lags behind our EST approach by a large margin. Furthermore, the baselines we adopt in the original paper follow those used in ESTAG, which are widely recognized and accepted within the community. This ensures that our comparisons are fair and credible. Moreover, both ESTAG and Eqmotion are strong baselines, and our method achieves significant improvements over them.
>
> **Table S7:  Performance between GNS, Equiformer, and EST on MD17.**
> | Method     | Aspirin      | Benzene      | Ethanol      | Malonaldehyde | Naphthalene  | Salicylic    | Toluene      | Uracil       |
> | :--------- | :----------- | :----------- | :----------- | :------------ | :----------- | :----------- | :----------- | :----------- |
> | GNS        | 23.545±0.001 | 2.53±0.002   | 15.793±0.001 | 17.878±0.001  | 12.768±0.006 | 22.498±0.001 | 12.109±0.001 | 21.299±0.001 |
> | Equiformer | 10.13        | 2.00         | 1.88         | 8.05          | 3.43         | 5.79         | 2.09         | 4.38         |
> | EST        | **2.196±0.075**  | **0.480±0.050**  | **0.940±0.001**  | **1.762±0.054**   | **0.988±0.016**  | **1.733±0.031**  | **1.002±0.063**  | **1.087±0.055**  |
>
> > **Weakness3: The model’s ability to generalize to systems with different numbers of components or interaction types is not convincingly demonstrated, which is a critical capability for physical simulation models.**
>
> Thank you for raising this critical question. To further validate our model's ability to generalize to systems with different numbers of components or interaction types, we conduct the following experiment. We select one molecule as the unseen test target at a time and combine the datasets of the remaining 7 molecules as the training set. Since the molecules in MD17 exhibit significant differences in atomic counts and chemical structures (interaction types), this setup rigorously tests the model's generalization capability across systems with varying component numbers and interaction types. We compare our EST model with the previous SOTA model, ESTAG, under this generalization setting (labeled as -generalize). The results are presented in Table S8 below. Our EST model outperforms ESTAG on all molecules in this generalization setting. On average, the error of EST-generalize (2.760) is significantly lower than that of ESTAG-generalize (3.527). This demonstrates that our architecture captures universal physical dynamic laws more effectively than previous methods and successfully generalizes to systems with different numbers of components.
>
> **Table S8:  Performance under generalization setting on MD17.**
> |                  | Aspirin     | Benzene     | Ethanol     | Malonaldehyde | Naphthalene | Salicylic   | Toluene     | Uracil      | Average |
> | :--------------- | :---------- | :---------- | :---------- | :------------ | :---------- | :---------- | :---------- | :---------- | :------ |
> | ESTAG            | 2.553±0.414 | 1.524±0.142 | 0.977±0.001 | 2.758±0.794   | 2.278±0.211 | 2.239±0.576 | 1.733±0.591 | 1.600±0.237 | 1.958   |
> | ESTAG-generalize | 5.037±0.838 | 2.080±0.177 | 2.763±0.085 | 5.823±0.195   | 3.099±0.390 | 2.791±0.094 | 3.515±0.370 | 3.114±0.179 | 3.527   |
> | EST              | 2.196±0.075 | 0.480±0.050 | 0.940±0.001 | 1.762±0.054   | 0.988±0.016 | 1.733±0.031 | 1.002±0.063 | 1.087±0.055 | 1.273   |
> | EST-generalize   | 3.747±0.383 | 1.685±0.188 | 2.613±0.181 | 5.756±0.262   | 1.578±0.021 | 1.952±0.124 | 2.607±0.059 | 2.148±0.154 | 2.760   |

---

> ### Author Response · Authors · 2025-11-24
> **Responses to Reviewer isjy (Part 3/3)**
>
> Finally, we extend our sincere gratitude to the reviewer for suggesting further clarifications and refined descriptions, as well as for the recommendation to test our method's generalization on systems with different numbers of components or interaction types. These additions have greatly enriched our experiments and enhanced the overall quality and clarity of the paper.
>
> [A] Equivariant Graph Hierarchy-Based Neural Networks, NIPS 2022.
>
> [B] Equivariant Spatio-Temporal Attentive Graph Networks to Simulate Physical Dynamics, NIPS 2023.
>
> [C] EqMotion: Equivariant Multi-agent Motion Prediction with Invariant Interaction Reasoning, ICCV 2023.
>
> [D] Equivariant Graph Neural Operator for Modeling 3D Dynamics, ICML 2024.
>
> [E] Equiformer: Equivariant Graph Attention Transformer for 3D Atomistic Graphs. ICLR 2023.

---

> ### Author Response · Authors · 2025-11-27
> **Looking forward to your responses**
>
> Dear Reviewer isjy,
>
> We sincerely appreciate the time and effort you have dedicated to reviewing our paper. We have made every effort to address your questions and concerns in detail, and have updated the manuscript accordingly (with changes highlighted in red). If you find our responses satisfactory, we would be grateful for your reconsideration of the score!
>
> Best regards,
>
> The Authors

---

### Official Review · Reviewer_Uo2G · 2025-10-27

**Soundness:** 2
**Presentation:** 2
**Contribution:** 2
**Rating:** 2
**Confidence:** 5

**Summary:**

The paper reframes physical dynamics simulation—traditionally treated as predicting continuous trajectories—as a next-graph prediction problem using a Transformer-based architecture.
It introduces the Equivariant Spatiotemporal Transformer (EST), a model that predicts future graph states autoregressively while preserving E(3) symmetries (rotations, reflections, translations).
This allows EST to simulate molecular, protein, and human motion dynamics with geometric and temporal consistency.

**Strengths:**

This reframing connects sequence modeling (from NLP) with geometric dynamics, allowing the use of autoregressive Transformers for temporal evolution in physical systems.

It provides a unifying perspective that bridges graph-based physics learning, trajectory modeling, and generative dynamics under a single formalism.

**Weaknesses:**

Limited Physical Grounding Beyond Equivariance
Autoregressive Rollout Still Accumulates Error

**Questions:**

Why “next-graph prediction”? and this method is not new.
How does EST compare to Hamiltonian or Lagrangian neural networks that explicitly encode energy and momentum conservation?
Can the model generalize to continuous-time prediction (e.g., irregular timesteps)?
Can model predict rapid change unseen?

---

> ### Author Response · Authors · 2025-11-24
> **Responses to Reviewer Uo2G (Part 1/3)**
>
> We are deeply grateful for the time and thoughtful effort you have dedicated to offering such detailed and constructive feedback. Your valuable suggestions have significantly enhanced both the scholarly rigor and presentation of our manuscript. Having carefully revised the paper to reflect your comments, we now address each point in detail below.
>
> > **Weakness1:  Limited Physical Grounding Beyond Equivariance Autoregressive Rollout Still Accumulates Error**
>
> Thank you for your constructive comments. **Regarding "limited physical grounding,"** in addition to basic equivariance, we specifically design the Temporal Difference Graph (TDG), which reflects physical insights. Instead of predicting absolute positions directly, we predict variations. In Appendix B.8.2, we provide a theoretical basis using Taylor expansion. This effectively introduces an inductive bias similar to numerical differential equation solvers like the Euler method.
>
> **Regarding "accumulates error,"** we acknowledge that error accumulation is inherent to any autoregressive model. However, the core contribution of this work is to mitigate and suppress the rate of accumulation rather than eliminate it entirely. Our TDG acts as a global graph that interacts with all historical frames in the decoder. This ensures that each prediction step is constrained and corrected by the entire historical trajectory, thereby suppressing the amplification of local errors. This interaction is visualized in the attention map in Figure 5 of Appendix F. Our experimental results further validate this. On the MD17 dataset (Tables 14, 15, and 16) and Motion Capture dataset (Table 11), the performance gap between EST and baselines (e.g., ESTAG) widens significantly as the rollout steps increase from 10 to 15 and 20. Additionally, Figure 4 in Appendix E.1 intuitively shows that while the MSE of many baseline models rises sharply with increasing steps, the curve for EST remains significantly flatter.
>
> > **Question1: Why “next-graph prediction”? and this method is not new.**
>
> Thank you for your question. **Regarding "Why next-graph prediction":** In the systems we study (e.g., molecules, proteins, human motion), the state is naturally represented as a graph where atoms or joints are nodes and bonds or connections are edges. Dynamics essentially describe the evolution of node attributes (specifically 3D coordinates) over time. Therefore, simulating dynamics is equivalent to predicting the graph's state at the next timestep. Furthermore, unlike traditional GNN methods (e.g., EGNN, ESTAG) that typically process fixed windows, we treat physical trajectories as sequence data, analogous to "Next-Token Prediction" in NLP. This perspective allows us to leverage the Transformer's ability to capture long-range dependencies for variable-length historical inputs. Hence, we call it "next-graph prediction".
>
> **Regarding "this method is not new":** The novelty of EST lies in its unique combination, architectural innovation, and specific application domain. EST is the first work to rigorously integrate $E(3)$-equivariance into a Transformer encoder-decoder architecture specifically for autoregressive physical simulation. Unlike architectures that use only an encoder or decoder, EST utilizes the encoder to compress historical trajectories and the decoder for autoregressive prediction. We also introduce the novel TDG, a learnable, dynamic temporal graph that interacts with all historical frames via attention in the decoder to capture global dynamics, a feature absent in prior works.

---

> ### Author Response · Authors · 2025-11-24
> **Responses to Reviewer Uo2G (Part 2/3)**
>
> > **Question2: How does EST compare to Hamiltonian or Lagrangian neural networks that explicitly encode energy and momentum conservation?**
>
> Thank you for your question. While Hamiltonian Neural Networks (HNN) and Lagrangian Neural Networks (LNN) theoretically guarantee energy conservation by learning the system's Hamiltonian or Lagrangian, fitting a complex potential energy surface in high-dimensional spaces (such as molecular and protein systems) proves relatively difficult. This strict mathematical form often leads to optimization challenges and sensitivity to noise, resulting in underfitting. In contrast, EST leverages $E(3)$ equivariance and the Transformer architecture; its advantage lies in its strong expressivity and flexibility. The Transformer architecture captures extremely complex spatiotemporal dependencies, while $E(3)$ equivariance ensures generalization across arbitrary coordinate systems.
>
> To further address your concern, we reproduce the HNN model on the standard MD17 benchmark and compare it with EST, as shown in Table S3. The results demonstrate that EST achieves significantly lower prediction errors than HNN across all molecular tasks, with performance improvements ranging from 3 to 7 times. This strongly suggests that for real-world, complex molecular dynamics data involving high-order interactions, the rigid physical constraints of HNN limit fitting capability. EST, by combining physical symmetry with the expressive power of Transformers, simulates the complex trajectories of microscopic particles more precisely while maintaining physical consistency.
>
> **Table S3:  Performance between HNN and EST on MD17.**
> |       | Aspirin       | Benzene       | Ethanol       | Malonaldehyde | Naphthalene   | Salicylic     | Toluene       | Uracil        |
> | :---- | :------------ | :------------ | :------------ | :------------ | :------------ | :------------ | :------------ | :------------ |
> | HNN   | 8.378±0.057   | 1.537±0.060   | 3.292±0.001   | 6.228±0.009   | 7.412±0.052   | 7.942±0.129   | 5.786±0.051   | 7.003±0.168   |
> | EST   | **2.196±0.075**   | **0.480±0.050**   | **0.940±0.001**   | **1.762±0.054**   | **0.988±0.016**   | **1.733±0.031**   | **1.002±0.063**   | **1.087±0.055**   |
>
> > **Question3: Can the model generalize to continuous-time prediction (e.g., irregular timesteps)?**
>
> Thank you for your constructive question. To verify whether our model generalizes to continuous-time prediction, we conduct an experiment where models are trained on MD17 with a fixed sampling interval ($\Delta t = 10$) but tested on random intervals sampled from the range $[1, 30]$ (e.g., intervals of 15, 5, or 20 frames). We compare EST with the previous SOTA model, ESTAG, and the results are presented in Table S4 (suffixed with -irregular). The data reveals that ESTAG suffers severe performance degradation under irregular timesteps; for instance, the error on Benzene spikes from 1.524 to 14.806 (a nearly 9-fold increase), indicating that traditional methods rely heavily on a fixed $\Delta t$ and fail to learn the continuous dynamics equation. In contrast, EST demonstrates remarkable stability. Despite the significant increase in task difficulty, EST maintains low error levels under irregular sampling, outperforming ESTAG across all molecules. This indicates that EST, through the TDG module's differential modeling and the Transformer's attention mechanism, adaptively adjusts prediction magnitudes based on implicit time spans, thereby naturally generalizing to continuous or irregular prediction tasks.
>
> **Table S4:  Performance under irregular timesteps on MD17.**
> |                 | Aspirin       | Benzene       | Ethanol       | Malonaldehyde | Naphthalene   | Salicylic     | Toluene       | Uracil        |
> | :-------------- | :------------ | :------------ | :------------ | :------------ | :------------ | :------------ | :------------ | :------------ |
> | ESTAG           | 2.553±0.414   | 1.524±0.142   | 0.977±0.001   | 2.758±0.794   | 2.278±0.211   | 2.239±0.576   | 1.733±0.591   | 1.600±0.237   |
> | ESTAG-irregular | 10.861±1.175  | 14.806±0.282  | 1.650±0.001   | 4.196±0.230   | 5.509±0.339   | 5.060±0.246   | 5.704±2.933   | 4.468±0.533   |
> | EST             | 2.196±0.075   | 0.480±0.050   | 0.940±0.001   | 1.762±0.054   | 0.988±0.016   | 1.733±0.031   | 1.002±0.063   | 1.087±0.055   |
> | EST-irregular   | 3.679±0.128   | 0.875±0.009   | 1.488±0.001   | 2.677±0.046   | 1.874±0.031   | 3.434±0.084   | 2.041±0.103   | 2.205±0.024   |

---

> ### Author Response · Authors · 2025-11-24
> **Responses to Reviewer Uo2G (Part 3/3)**
>
> > **Question4: Can model predict rapid change unseen?**
>
> Nice suggestion! To validate the model's ability to predict unseen rapid changes, we design an experiment where models are trained on data with a fixed interval of $\Delta t = 10$ but tested on a larger interval of $\Delta t = 15$. This represents a 50% increase in particle displacement within the same physical time, constituting a "rapid change" unseen during training. We compare EST against ESTAG, with results shown in Table S5 (suffixed with rapid-change). As indicated in the table, ESTAG exhibits catastrophic failure when facing these unseen rapid changes; for example, the error on Benzene increases nearly 15-fold (from 1.524 to 23.808), and the error on Naphthalene increases nearly 7-fold. This suggests that traditional models severely overfit the motion magnitude observed during training. Conversely, EST demonstrates exceptional robustness. On Benzene, despite the 50% increase in magnitude, the error only rises to 1.510, which outperforms ESTAG's performance on the standard test set. On complex molecules like Naphthalene and Salicylic, EST's error is only 1/5 to 1/3 of ESTAG's. This capability stems from EST's modeling of differences and the long-range attention mechanism; the model learns trends of change rather than memorizing positions, enabling effective extrapolation when the magnitude ($\Delta t$) changes.
>
> **Table S5:  Performance under rapid change on MD17.**
> |                    | Aspirin       | Benzene       | Ethanol       | Malonaldehyde | Naphthalene   | Salicylic     | Toluene       | Uracil        |
> | :----------------- | :------------ | :------------ | :------------ | :------------ | :------------ | :------------ | :------------ | :------------ |
> | ESTAG              | 2.553±0.414   | 1.524±0.142   | 0.977±0.001   | 2.758±0.794   | 2.278±0.211   | 2.239±0.576   | 1.733±0.591   | 1.600±0.237   |
> | ESTAG-rapid-change | 13.887±1.615  | 23.808±1.688  | 3.947±0.001   | 7.626±0.279   | 17.884±0.889  | 15.042±0.746  | 6.932±0.061   | 7.690±1.092   |
> | EST                | 2.196±0.075   | 0.480±0.050   | 0.940±0.001   | 1.762±0.054   | 0.988±0.016   | 1.733±0.031   | 1.002±0.063   | 1.087±0.055   |
> | EST-rapid-change   | 6.547±0.104   | 1.510±0.168   | 4.007±0.001   | 5.342±0.083   | 3.727±0.101   | 6.161±0.490   | 4.272±0.363   | 4.271±0.141   |
>
> Finally, we sincerely thank the reviewer for these constructive suggestions, particularly the recommendation to compare our method with HNN and to evaluate its generalization capabilities under irregular timesteps and unseen scenarios with rapid changes. These supplementary experiments have made our comparative analysis more comprehensive, further highlighted the advantages of our approach, and significantly improved the overall quality of our manuscript.

---

> ### Author Response · Authors · 2025-11-27
> **Looking forward to your responses**
>
> Dear Reviewer Uo2G,
>
> We sincerely appreciate the time and effort you have dedicated to reviewing our paper. We have made every effort to address your questions and concerns in detail, and have updated the manuscript accordingly (with changes highlighted in red). If you find our responses satisfactory, we would be grateful for your reconsideration of the score!
>
> Best regards,
>
> The Authors

---

### Official Review · Reviewer_Vq7q · 2025-11-05

**Soundness:** 3
**Presentation:** 3
**Contribution:** 3
**Rating:** 6
**Confidence:** 4

**Summary:**

The paper proposes an attention-based encoder-decoder for molecule dynamic problems. I think the key part is Equivariant Spatiotemporal Attention. I think the overall design is not complicated, but useful for the future work.

**Strengths:**

The design is simple and easy to follow. And the experiments are promising. The overall design is close to modern NLP (gpt-like), but also considers the property of the molecules.

**Weaknesses:**

1. It looks like the model can not handle large molecules due to the token number (each node represents one token, not efficient.)
2. The model seems not to consider the connection for the 3d molecules.
3. The title is not proper. The proposed method can only handle dynamic molecule problem, not general physical dynamics.

**Questions:**

1. What is the average node number for the dataset used?
2.  10 rollout steps looks very small, is it possible to expand it?

---

> ### Author Response · Authors · 2025-11-24
> **Responses to Reviewer Vq7q (Part 1/2)**
>
> We sincerely appreciate the time and effort you have devoted to providing detailed and constructive feedback. Your insightful comments have been invaluable in improving both the technical quality and clarity of our manuscript. We have carefully revised our paper to incorporate your suggestions. Below, we address each of your points individually.
>
> > **Weakness1: It looks like the model can not handle large molecules due to the token number (each node represents one token, not efficient.)**
>
> Thank you for your question. We specifically conducted experiments on Protein Dynamics in Section 4.3 of the main text. Proteins are typical large molecular systems, and in this task, we treated each residue as a single node by constructing a 4-channel equivariant 3D coordinate for the four backbone atoms (N, Cα, C, O). By following a similar approach, our model is capable of handling large molecules efficiently. Furthermore, in Appendix E.9, we present experimental results for all-atom protein dynamics. The results are shown in Table S1, demonstrating that the model maintains excellent performance in this setting. We also propose in Appendix E.8 that virtual nodes technology [A] could be employed in the future to further process extremely large-scale systems.
>
> **Table S1: All-atom protein dynamics.**
> |                           | EGNN  | ESTAG | EST   |
> | :------------------------ | :---- | :---- | :---- |
> | All-atom protein dynamics | 2.531 | 2.533 | **2.430** |
>
> > **Weakness2: The model seems not to consider the connection for the 3d molecules.**
>
> Thank you for your detailed observation. The decision not to explicitly use edge features (connection) is an experimentally verified choice that does not hinder the model's ability to capture geometric connections. As mentioned in Appendix B.6, we attempted to incorporate edge features (such as bond types) but find that they yielded negligible performance gains; therefore, to reduce model complexity and maintain generality, we chose to remove them. Regarding angular features within connections, as noted in Section 3.3 of PaiNN [B], if distance information is introduced at every layer, the stacked layers can implicitly model angular information within the network, meaning our model effectively learns angular geometries. Additionally, as detailed in Appendices C.2 and C.4, although we do not explicitly use edge features, we dynamically construct graph structures based on distance thresholds. This means that physically connected (bonded) atoms are naturally included as neighbor nodes for message passing, thereby effectively capturing connection information.
>
> > **Weakness3: The title is not proper. The proposed method can only handle dynamic molecule problem, not general physical dynamics.**
>
> Thank you for your suggestion. In our paper, in addition to experiments on the MD17 dataset for molecular dynamics and the Adk equilibrium trajectory dataset for protein dynamics, we also conducted experiments on the Motion Capture dataset, which describes 3D trajectories of human motion. The Motion Capture dataset involves complex macroscopic physical behaviors such as walking and playing basketball, which differ significantly from microscopic molecular dynamics. Our model achieves SOTA performance on these tasks as well. The results are shown in Table 2 in the main text, demonstrating its generalization capability for general physical systems. Therefore, we collectively refer to these tasks as physical dynamics, following the terminology used in the prior work ESTAG [C] in this field.
>
> > **Question1: What is the average node number for the dataset used?**
>
> Thank you for your question. As mentioned in Appendix C, in the MD17 dataset, each graph contains up to 13 nodes. In the Motion Capture dataset, each graph contains up to 31 nodes. In the Protein dataset, each graph contains 213 nodes.

---

> ### Author Response · Authors · 2025-11-24
> **Responses to Reviewer Vq7q (Part 2/2)**
>
> > **Question2: 10 rollout steps looks very small, is it possible to expand it?**
>
> Thank you for your valuable comments. Conducting long-term rollouts for physical dynamics simulations is inherently a challenging task. Considering the characteristics of the datasets, such as the highly oscillatory dynamics in MD17, a rollout length of 20 steps is already an effective and reasonable choice for evaluating model performance. For reference, previous methods such as ESTAG use only 10-step rollouts.
>
> To further address your concern, Table S2 reports 100-step rollout results on MD17, comparing our method with ESTAG. The results show that our method still consistently outperforms ESTAG. However, after 100 rollout steps, the model's MSE deteriorates from 1e-5 to 1e-3 in magnitude. This highlights that, due to the dramatic dynamics of MD17, achieving stable rollouts over 100 steps remains extremely challenging.
>
> **Table S2: Predicted MSE ($\times 10^{-3}$) for 100-step rollout results on MD17.**
> |       | R=1   | R=25  | R=50  | R=75  | R=100 |
> | :---- | :---- | :---- | :---- | :---- | :---- |
> | ESTAG | 0.090 | 5.969 | 6.637 | 6.378 | 6.406 |
> | EST   | **0.078** | **4.181** | **5.343** | **6.163** | **5.993** |
>
> Finally, we express our sincere gratitude to the reviewer for the constructive comments and the requirement to expand the rollout step experiments. Incorporating these comments and results has not only made our work more holistic but also substantially enhanced the rigorousness and overall quality of our presentation.
>
> [A] Improving equivariant graph neural networks on large geometric graphs via virtual nodes learning. ICML 2024.
>
> [B] Equivariant message passing for the prediction of tensorial properties and molecular spectra. ICML 2021
>
> [C] Equivariant Spatio-Temporal Attentive Graph Networks to Simulate Physical Dynamics. NIPS 2023.

---

> ### Author Response · Authors · 2025-11-27
> **Looking forward to your responses**
>
> Dear Reviewer Vq7q,
>
> We sincerely appreciate the time and effort you have dedicated to reviewing our paper. We have made every effort to address your questions and concerns in detail, and have updated the manuscript accordingly (with changes highlighted in red). If you find our responses satisfactory, we would be grateful for your reconsideration of the score!
>
> Best regards,
>
> The Authors

---

### Author Response · Authors · 2025-12-03
**Rebuttal Acknowledgment (Part 1/2)**

Dear Area Chair,

We sincerely appreciate your invaluable support and guidance throughout the review and discussion process. We would also like to express our deepest gratitude to the reviewers for their diligent efforts, insightful feedback, and constructive suggestions. To facilitate your evaluation of our paper, we provide a summary of the paper's content and the rebuttal below:

> **Overview of our paper:**

Our paper explores autoregressive prediction for physical dynamics simulation, innovatively combining Transformer architecture with equivariance for such tasks while proposing the Temporal Difference Graph (TDG) to reduce accumulated errors in autoregressive prediction. Extensive experiments demonstrate that our model achieves state-of-the-art performance across multi-scale physical systems, while also exhibiting excellent generalization capability and robustness when confronted with unseen scenarios.

> **Overview of the rebuttal:**

We are pleased that the **innovation and significance** of our approach have been recognized by the reviewers, who described it as:
- *"useful for the future work"* (Reviewer Vq7q),
- *"provides a unifying perspective that bridges graph-based physics learning... under a single formalism"* (Reviewer Uo2G),
- *"conceptually interesting, and could potentially offer a unified framework for structured dynamical modeling"* (Reviewer isjy),
- *"a creative and promising approach to the critical problem of error accumulation"* (Reviewer HooZ).

Owing to the effective design of our method, EST demonstrates **versatility and strong empirical performance.** This advantage has been highlighted as:
- *"experiments are promising"* (Reviewer Vq7q),
- *"simulate molecular, protein, and human motion dynamics with geometric and temporal consistency"* (Reviewer Uo2G),
- *"well-suited to modeling the symmetries inherent in physical systems"* (Reviewer isjy),
- *"The experiments are extensive, spanning three distinct and relevant physical scales, ... convincingly demonstrates the model's versatility and generalizability"* (Reviewer HooZ).

Since we did not receive further responses from the four reviewers before the discussion period concluded on November 28/29, we were unable to engage in more in-depth discussions with them. Nevertheless, we believe we have comprehensively addressed all major concerns as follows: (1) provided detailed clarifications for all questions raised by the reviewers regarding the manuscript, (2) meticulously incorporated their suggestions into the revised version, with all modifications highlighted in red, and (3) conducted extensive additional experiments to address every methodological and experimental concern raised. Below, we provide a concise summary of our responses to each reviewer's comments. For full details, please refer to our point-by-point response.

**Response to Reviewer Vq7q**
- Handling large molecules & efficiency: We clarify our efficient 4-channel encoding strategy and **Table S1** presents new experiments on **all-atom protein dynamics**. These results confirm that the model scales effectively to large systems (e.g., 213 nodes per graph). Furthermore, we discussed the potential of virtual nodes for extreme scales in **Appendix E.8**.
- Lack of explicit edge features: We justify this design choice, as experiments showed that explicit edges yielded negligible gains. Instead, we rely on **dynamic graph** construction and stacked layers to implicitly capture angular geometry and physical connections without redundancy.
- Justification of the title scope: We justify the scope by highlighting our SOTA performance on the **macroscopic Motion Capture dataset** (e.g., human walking). This confirms that the model generalizes well to diverse physical behaviors beyond microscopic molecular dynamics.
- Average number of nodes: We provide the specific average node number for all utilized datasets.
- Request to expand rollout steps: We address this by conducting **new 100-step rollout experiments (Table S2).** Despite the chaotic nature of long-term predictions, our model consistently outperforms the baseline (ESTAG) in these extended horizons.

---

> ### Author Response · Authors · 2025-12-03
> **Rebuttal Acknowledgment (Part 2/2)**
>
> **Response to Reviewer Uo2G**
> - Physical grounding and error accumulation: **Regarding "limited physical grounding:"** In addition to basic equivariance, we specifically designed the Temporal Difference Graph (TDG), which reflects physical insights. In Appendix B.8.2, we also provide a theoretical basis using Taylor expansion. **Regarding "error accumulation:"** We acknowledge that error accumulation is inherent to any autoregressive model. However, the core contribution of this work is to **mitigate and suppress the rate of accumulation** rather than eliminate it entirely. The experimental results on the MD17 dataset (Tables 14, 15, and 16), the Motion Capture (Table 11), and Figure 4 in Appendix E.1 provide robust support for this point.
> - "Next-graph prediction" and novelty: **Regarding "next-graph prediction":** We clarified the "Next-Graph Prediction" concept, drawing a parallel to "Next-Token Prediction" in NLP. **Regarding "the novelty":** We underscore the architectural novelty: EST is the first E(3)-equivariant Transformer encoder-decoder framework designed specifically for physical dynamics simulation.
> - Comparison with Hamiltonian/Lagrangian NNs: In **Table S3**, we compare our method with Hamiltonian Neural Networks. Results show our model achieves 3 to 7 times lower error when compared to HNNs.
> - Generalization to continuous-time (irregular timesteps): **Table S4** presents experiments on irregular time intervals (**train on $\Delta t$ = 10, test on $\Delta t$ randomly sampled from the range [1, 30]**). Results in Table S4 demonstrate our model remains robust and stable while the baseline suffers severe performance degradation.
> - Predicting unseen rapid changes: **Table S5** demonstrates our model's performance under rapid changes (**train on $\Delta t$ = 10, test on $\Delta t$  = 15**). The results in Table S5 confirm our model maintains low error rates while baseline exhibit catastrophic failure (up to 15-fold error increase).
>
> **Response to Reviewer isjy**
> - Underdeveloped technical details (Graph Construction & Feature Evolution): **Regarding "Graph Construction"**: **Table S6** briefly illustrates the feature information of the graphs for each dataset. For more detailed graph construction, we provide comprehensive descriptions in Appendices C.2, C.3, and C.4. **Regarding "Feature Evolution":** We introduce this process in detail in Section 3.2, and provide a **brief overview** of how node and edge features evolve in our response below.
> - Limited benchmarks and missing baselines (e.g., GNS): The three datasets listed in our paper are widely recognized and popular in this field. **Table S7** strengthens our evaluation by including **comparisons with strong baselines like GNS and Equiformer**. Results demonstrate our method significantly outperforms these established approaches on standard benchmarks.
> - Generalization to systems with different components: **Table S8** validates this ability through new cross-system generalization experiments (**train on a subset of molecules and test on other unseen molecules**). The results in Table S8 demonstrate our model achieves **lower error across all molecules**, and attains a significantly lower mean error (2.760) compared to the SOTA baseline ESTAG (3.527).
>
> **Response to Reviewer HooZ**
> - Superiority of TDG over simple velocity prediction: We clarify that unlike local velocity methods (e.g., GNS), our TDG captures global trends by attending to all historical frames. **Table S9** demonstrates this superiority through comparative experiments with GNS, where our method **achieves an order-of-magnitude lower error** (e.g., 2.196 vs. 23.545 on Aspirin).
> - Computational cost and runtime analysis: We first explain the context for the term "negligible". We then revise the manuscript to transparently discuss the trade-off: while slower than some baselines, our model delivers a **qualitative breakthrough in accuracy (e.g., 68% error reduction on Benzene).** We also outline future acceleration strategies, such as flash attention and virtual nodes.
> - Rationale for predicting differences (TDG) vs. next frame: We explain that predicting differences is superior as it avoids the inefficiency of learning identity mappings and **simplifies optimization by targeting sparse, near-zero fluctuations rather than large absolute coordinates.** This is empirically supported by the **"w/o TDG" ablation study (Table 4),** where removing differential modeling leads to a significant performance drop.
> - Clarification on Eq. (4) (Mean Reduction): We clarify that the mean reduction term is computed **only once** based on the initial input trajectory.
>
> Once again, we sincerely appreciate your time and support. Your leadership in overseeing the review process has been invaluable. We also extend our heartfelt thanks to the reviewers for their dedication; their feedback has significantly improved the quality of our manuscript.
>
> Best regards,
>
> The Authors of Submission 9097

---

### Meta-Review · Area_Chair_ZZkT · 2026-01-02

**Summary:**

This paper reformulates physical dynamics simulation as autoregressive spatiotemporal graph prediction. The approach is technically sound and well aligned with symmetry principles. The rebuttal effectively clarifies the motivation, design choices, and advantages over existing GNN-based methods through additional experiments. Overall, the contribution is solid. However, compared with other strong submission, I lean toward reject, slightly below the acceptance threshold.

**Reviewer Concerns:**

The rebuttal effectively addresses the core concerns raised by all four reviewers. Concerns regarding physical grounding and error accumulation (Reviewer Uo2G) are convincingly resolved through the introduction and theoretical justification of the Temporal Difference Graph, supported by extensive new experiments across long rollouts, irregular timesteps, and rapid-change regimes. Questions about novelty and positioning are clarified by formalizing “next-graph prediction” and by adding strong comparisons with Hamiltonian, Lagrangian, and graph-based baselines (e.g., HNN, GNS, Equiformer). Evaluation gaps noted by Reviewers Vq7q and isjy, including scalability, missing baselines, limited benchmarks, and generalization across systems, are comprehensively addressed with additional datasets, experiments, and ablations. Concerns about computational cost and the advantage of TDG over simpler formulations (Reviewer HooZ) are also directly discussed and empirically validated. Overall, no major reviewer concerns appear outstanding after the rebuttal.

**Reviewer Scores:**

**Reviewer Vq7q (original score 6 → 6)**

The reviewer raised concerns regarding scalability to large molecular systems, the lack of explicit edge features, the scope implied by the paper title, and the rollout length used in evaluation. The rebuttal provides clear empirical evidence and detailed justification addressing each point, including experiments on protein dynamics, analysis of implicit geometric modeling, and additional long-horizon rollout results. These responses largely confirm the soundness of the original design choices rather than introducing fundamentally new evidence. As a result, the reviewer would likely maintain their original assessment, keeping the score at 6.

**Reviewer isjy (original score 4 → 6)**

The reviewer initially expressed concerns about insufficient technical detail, limited experimental benchmarks, missing comparisons to strong simulators, and unclear generalization capability. The rebuttal substantially clarifies the mathematical and architectural details of graph construction, adds explicit references to appendices with formal definitions, and demonstrates that strong baselines such as GNS and Equiformer were already evaluated and are outperformed. While generalization across fundamentally different interaction regimes remains only partially explored, the added explanations and empirical evidence significantly strengthen the paper’s rigor and credibility. With these clarifications, the reviewer would likely revise their assessment upward, increasing the score to 6.

**Reviewer HooZ (original score 6 → 6)**

The reviewer raised technically nuanced concerns regarding the motivation and superiority of the TDG design, the computational cost trade-offs, the rationale for difference-based prediction, and the clarity of certain mathematical definitions. The rebuttal provides thorough and convincing explanations, including clear conceptual motivation, direct empirical comparisons with velocity-based baselines (e.g., GNS), supporting ablation evidence, and explicit clarification of the rollout formulation. While these responses substantially strengthen the paper’s clarity and justification, they primarily reinforce rather than fundamentally change the reviewer’s original positive assessment. As a result, the reviewer would likely maintain their score at 6, reflecting continued confidence in the technical soundness and contribution of the work.

**Reviewer Uo2G (original score 2 → 4)**

The reviewer initially expressed concerns about limited physical grounding, error accumulation in autoregressive rollout, the novelty of “next-graph prediction,” and the lack of comparison with physically motivated models and challenging generalization settings. The rebuttal directly addresses these points by clarifying the physical inductive bias introduced by the Temporal Difference Graph (TDG), providing theoretical justification and extensive empirical evidence that error accumulation is substantially mitigated over long rollouts. The authors also clearly articulate the novelty of framing dynamics as next-graph prediction with an equivariant Transformer and introduce new comparisons with Hamiltonian Neural Networks, as well as additional experiments on irregular timesteps and unseen rapid changes. These clarifications and added results substantially strengthen the paper’s physical motivation, novelty, and empirical credibility, likely leading the reviewer to raise their score to 4.

---

### Decision · Program_Chairs · 2026-01-26

Reject